# An international observational study to assess the impact of the Omicron variant emergence on the clinical epidemiology of COVID-19 in hospitalised patients

Bronner P Gonçalves[1]*, Matthew Hall[2], Waasila Jassat[3], Valeria Balan[1], Srinivas Murthy[4], Christiana Kartsonaki[5], Malcolm G Semple[6,7], Amanda Rojek[1,8,9], Joaquín Baruch[1], Luis Felipe Reyes[1,10,11], Abhishek Dasgupta[12,13], Jake Dunning[1], Barbara Wanjiru Citarella[1], Mark Pritchard[1], Alejandro Martín-Quiros[14], Uluhan Sili[15], J Kenneth Baillie[16,17], Diptesh Aryal[18], Yaseen Arabi[19], Aasiyah Rashan[20], Andrea Angheben[21], Janice Caoili[22], François Martin Carrier[23,24,25,26], Ewen M Harrison[27], Joan Gómez-Junyent[28], Claudia Figueiredo-Mello[29], James Joshua Douglas[30], Mohd Basri Mat Nor[31], Yock Ping Chow[32], Xin Ci Wong[33], Silvia Bertagnolio[34], Soe Soe Thwin[34], Anca Streinu-Cercel[35,36], Leonardo Salazar[37], Asgar Rishu[38], Rajavardhan Rangappa[39], David SY Ong[40], Madiha Hashmi[41], Gail Carson[1], Janet Diaz[34], Rob Fowler[42], Moritz UG Kraemer[13,43], Evert-Jan Wils[44], Peter Horby[1], Laura Merson[1,45], Piero L Olliaro[1], ISARIC Clinical Characterisation Group

*For correspondence: bronner.goncalves@ndm.ox.ac.uk

Group author details: ISARIC Clinical Characterisation Group See page 15

[1]ISARIC, Pandemic Sciences Institute, University of Oxford, Oxford, United Kingdom; [2]Big Data Institute, Nuffield Department of Medicine, University of Oxford, Oxford, United Kingdom; [3]National Institute for Communicable Diseases, South Africa; Right to Care, Johannesburg, South Africa; [4]Faculty of Medicine, University of British Columbia, Vancouver, Canada; [5]MRC Population Health Research Unit, Clinical Trials Service Unit and Epidemiological Studies Unit, Nuffield Department of Population Health, University of Oxford, Oxford, United Kingdom; [6]Institute of Infection, Veterinary and Ecological Sciences, Faculty of Health and Life Sciences, University of Liverpool, Liverpool, United Kingdom; [7]Respiratory Medicine, Alder Hey Children's Hospital, University of Liverpool, Liverpool, United Kingdom; [8]Royal Melbourne Hospital, Melbourne, Australia; [9]Centre for Integrated Critical Care, University of Melbourne, Melbourne, Australia; [10]Universidad de La Sabana, Chia, Colombia; [11]Clinica Universidad de La Sabana, Chia, Colombia; [12]Department of Computer Science, University of Oxford, Oxford, United Kingdom; [13]Department of Biology, University of Oxford, Oxford, United Kingdom; [14]Emergency Department. Hospital Universitario La Paz – IdiPAZ, Madrid, Spain; [15]Department of Infectious Diseases and Clinical Microbiology, School of Medicine, Marmara University, Istanbul, Turkey; [16]Roslin Institute, University of Edinburgh, Edinburgh, United Kingdom; [17]Intensive Care Unit, Royal Infirmary of Edinburgh, Edinburgh, United Kingdom; [18]Critical Care and Anesthesia, Nepal Mediciti Hospital, Lalitpur, Nepal; [19]King Abdullah International Medical Research Center and King Saud Bin Abdulaziz University for Health Sciences, Riyadh, Saudi Arabia; [20]Network for Improving Critical care Systems and Training, Colombo, Sri Lanka; [21]Department of Infectious, Tropical Diseases and Microbiology (DITM), IRCCS Sacro Cuore Don Calabria Hospital,

Negrar di Valpolicella, Verona, Italy; [22]Makati Medical Center, Makati City, Makati, Philippines; [23]Department of Anesthesiology, Centre hospitalier de l'Université de Montréal, Montréal, Canada; [24]Department of Medicine, Critical Care Division, Centre hospitalier de l'Université de Montréal, Montréal, Canada; [25]Carrefour de l'innovation et santé des populations, Centre de recherche du Centre hospitalier de l'Université de Montréal (CRCHUM), Montréal, Canada; [26]Department of Anesthesiology and Pain Medicine, Université de Montréal, Montréal, Canada; [27]Centre for Medical Informatics, The University of Edinburgh, Usher Institute of Population Health Sciences and Informatics, Edinburgh, United Kingdom; [28]Department of Infectious Diseases, Hospital del Mar, Infectious Pathology and Antimicrobial Research Group (IPAR), Institut Hospital del Mar d'Investigacions Mèdiques (IMIM), Universitat Autònoma de Barcelona (UAB), CEXS-Universitat Pompeu Fabra, Barcelona, Spain; [29]Instituto de Infectologia Emílio Ribas, São Paulo, Brazil; [30]Lions Gate Hospital, North Vancouver, Canada; [31]International Islamic University Malaysia, Selangor, Malaysia; [32]Clinical Research Centre, Sunway Medical Centre, Selangor Darul Ehsan, Selangor, Malaysia; [33]Digital Health Research and Innovation Unit, Institute for Clinical Research, National Institutes of Health (NIH), Selangor, Malaysia; [34]World Health Organization, Genève, Switzerland; [35]Carol Davila University of Medicine and Pharmacy, Bucharest, Romania; [36]National Institute for Infectious Diseases "Prof. Dr. Matei Bals", Bucharest, Romania; [37]Fundación Cardiovascular de Colombia, Santander, Colombia; [38]Department of Critical Care Medicine, Sunnybrook Health Sciences Centre, Toronto, Canada; [39]Department of Critical Care Medicine, Manipal Hospital Whitefield, Bengaluru, India; [40]Department of Medical Microbiology and Infection Control, Franciscus Gasthuis & Vlietland, Rotterdam, Netherlands; [41]Critical Care Asia and Ziauddin University, Karachi, Pakistan; [42]Department of Critical Care Medicine, Sunnybrook Health Sciences Centre, Toronto, Canada; [43]Pandemic Sciences Institute, University of Oxford, Oxford, United Kingdom; [44]Department of Intensive Care, Franciscus Gasthuis & Vlietland, Rotterdam, Netherlands; [45]Infectious Diseases Data Observatory, Centre for Tropical Medicine and Global Health, University of Oxford, Oxford, United Kingdom

## Abstract

**Background:** Whilst timely clinical characterisation of infections caused by novel SARS-CoV-2 variants is necessary for evidence-based policy response, individual-level data on infecting variants are typically only available for a minority of patients and settings.

**Methods:** Here, we propose an innovative approach to study changes in COVID-19 hospital presentation and outcomes after the Omicron variant emergence using publicly available population-level data on variant relative frequency to infer SARS-CoV-2 variants likely responsible for clinical cases. We apply this method to data collected by a large international clinical consortium before and after the emergence of the Omicron variant in different countries.

**Results:** Our analysis, that includes more than 100,000 patients from 28 countries, suggests that in many settings patients hospitalised with Omicron variant infection less often presented with commonly reported symptoms compared to patients infected with pre-Omicron variants. Patients with COVID-19 admitted to hospital after Omicron variant emergence had lower mortality compared to patients admitted during the period when Omicron variant was responsible for only a minority of infections (odds ratio in a mixed-effects logistic regression adjusted for likely confounders, 0.67 [95% confidence interval 0.61–0.75]). Qualitatively similar findings were observed in sensitivity analyses with different assumptions on population-level Omicron variant relative frequencies, and in analyses using available individual-level data on infecting variant for a subset of the study population.

**Conclusions:** Although clinical studies with matching viral genomic information should remain a priority, our approach combining publicly available data on variant frequency and a multi-country

clinical characterisation dataset with more than 100,000 records allowed analysis of data from a wide range of settings and novel insights on real-world heterogeneity of COVID-19 presentation and clinical outcome.

**Funding:** Bronner P. Gonçalves, Peter Horby, Gail Carson, Piero L. Olliaro, Valeria Balan, Barbara Wanjiru Citarella, and research costs were supported by the UK Foreign, Commonwealth and Development Office (FCDO) and Wellcome [215091/Z/18/Z, 222410/Z/21/Z, 225288/Z/22/Z]; and Janice Caoili and Madiha Hashmi were supported by the UK FCDO and Wellcome [222048/Z/20/Z]. Peter Horby, Gail Carson, Piero L. Olliaro, Kalynn Kennon and Joaquin Baruch were supported by the Bill & Melinda Gates Foundation [OPP1209135]; Laura Merson was supported by University of Oxford's COVID-19 Research Response Fund - with thanks to its donors for their philanthropic support. Matthew Hall was supported by a Li Ka Shing Foundation award to Christophe Fraser. Moritz U.G. Kraemer was supported by the Branco Weiss Fellowship, Google.org, the Oxford Martin School, the Rockefeller Foundation, and the European Union Horizon 2020 project MOOD (#874850). The contents of this publication are the sole responsibility of the authors and do not necessarily reflect the views of the European Commission. Contributions from Srinivas Murthy, Asgar Rishu, Rob Fowler, James Joshua Douglas, François Martin Carrier were supported by CIHR Coronavirus Rapid Research Funding Opportunity OV2170359 and coordinated out of Sunnybrook Research Institute. Contributions from Evert-Jan Wils and David S.Y. Ong were supported by a grant from foundation Bevordering Onderzoek Franciscus; and Andrea Angheben by the Italian Ministry of Health "Fondi Ricerca corrente–L1P6" to IRCCS Ospedale Sacro Cuore–Don Calabria. The data contributions of J.Kenneth Baillie, Malcolm G. Semple, and Ewen M. Harrison were supported by grants from the National Institute for Health Research (NIHR; award CO-CIN-01), the Medical Research Council (MRC; grant MC_PC_19059), and by the NIHR Health Protection Research Unit (HPRU) in Emerging and Zoonotic Infections at University of Liverpool in partnership with Public Health England (PHE) (award 200907), NIHR HPRU in Respiratory Infections at Imperial College London with PHE (award 200927), Liverpool Experimental Cancer Medicine Centre (grant C18616/A25153), NIHR Biomedical Research Centre at Imperial College London (award IS-BRC-1215-20013), and NIHR Clinical Research Network providing infrastructure support. All funders of the ISARIC Clinical Characterisation Group are listed in the appendix.

## Editor's evaluation

This manuscript compares COVID-19 mortality during the pre-Omicron and Omicron emergence periods in several countries. It finds evidence suggesting the Omicron variant was associated with lower mortality than previous dominant variants in multiple countries, though other factors than changing variant virulence might explain these observations, as discussed by the authors. This paper will be of interest to infectious disease scientists both for its content and its methods, as it validates that population-level variant frequency can be a good proxy for individual-level variant data to derive insights on variant biology with population data.

## Introduction

The emergence of novel SARS-CoV-2 variants represents a threat to the long-term control of COVID-19 (*Fontanet et al., 2021*). Whilst efforts to develop vaccines that protect against severe disease have been successful (*Polack et al., 2020*; *Voysey et al., 2021*; *Baden et al., 2021*), mutations in the viral genome that lead to ability to escape immunity, and increased transmissibility and/or clinical severity, either via intrinsic virulence or reduced vaccine effectiveness (*Lopez Bernal et al., 2021*), have the potential to cause substantial disease burden despite high vaccine coverage in many countries (*Our World in Data, 2021*).

These concerns motivated the prompt reporting, initially from South Africa (*Wolter et al., 2021*; *World Health Organization, 2022*), of clinical characteristics of infection with the Omicron variant only weeks after its emergence (*Wolter et al., 2022*; *Ulloa et al., 2022*; *Veneti et al., 2022*), which provided key information for risk assessment and health policies worldwide. Early data from South Africa showed reduced severity of Omicron lineage BA.1 and similar results were reported in the United Kingdom and the United States (*Wolter et al., 2022*; *Lewnard et al., 2022*; *Nyberg et al.,*

*2022*). However, the impact, in terms of clinical consequences (i.e. disease severity), of new variants has been shown to be context-specific, due to regional differences in disease epidemiology, including local circulation of previous variants and their cumulative incidences, variable vaccination coverages, and heterogeneity in population-level frequencies of risk factors (e.g. frequency of comorbidities) for severe disease and mortality. For this reason, international studies with standardised protocols are necessary to allow comparative assessments across different countries and epidemiological contexts.

To understand the impact of the emergence of the Omicron variant of SARS-CoV-2 on the clinical epidemiology of COVID-19 at the global level, in this study, we report multi-country data, from all six World Health Organization regions, on clinical characteristics and outcomes of Omicron variant infections in hospitalised patients and compare with infections in patients admitted with other SARS-CoV-2 variants. For that, we use publicly available population-level data on relative frequencies of the Omicron variant to determine periods when infections were likely to be caused by Omicron versus other variants/lineages and compare infections descriptively and using multivariable statistical models. In addition, we present an analysis that only includes patients with individual-level data on the infecting variant and paired clinical information.

## Methods

### ISARIC clinical characterisation protocol

Analyses presented in this manuscript use the ISARIC (International Severe Acute Respiratory and Emerging Infections Consortium) COVID-19 database, which includes prospectively collected data from countries where ISARIC partner institutions are located (see a global map of all ISARIC partner institutions here https://isaric.org/about-us/membership/). A full description of the data collection protocol and database can be found here https://isaric.org/research/covid-19-clinical-research-resources/. In short, data collection for this initiative was standardised, using the ISARIC case report forms, and pivoted into pandemic mode in January 2020 to enable rapid characterisation of the clinical presentation and severity of COVID-19. After the emergence of the Omicron variant, first reported in November 2021 (*Viana et al., 2022*), a call was launched to encourage international investigators partnering with ISARIC to rapidly share data on patients with confirmed or suspected COVID-19 to describe the clinical characteristics of Omicron variant infection in different settings; recommendations on possible hospitalised population sampling approaches were shared. Patients admitted to hospital from 1st October 2021 to 28th February 2022 were included in this analysis. More information on ISARIC can be found in *ISARIC Clinical Characterisation Group, 2021*; *Hall et al., 2021*; *Reyes et al., 2022*.

### Population-level SARS-CoV-2 variant data

Two statistical analysis plans (SAPs) were developed in December 2021 with approaches to be used in the characterisation of Omicron variant infection. Analyses described in the first SAP required individual-level data on the clinical presentation and paired data on the variant causing the infection. In the second SAP, we used population-level frequencies of SARS-CoV-2 lineages to infer individual infecting variant during different time periods as Omicron or non-Omicron variants (*Figure 1*). Since individual-level data on the infecting variant were limited to a few countries, these data are presented for comparison with the analysis performed using population-level variant data.

For the analysis that required information on population-level variant frequency, for countries contributing clinical data to this analysis, data from the Global Initiative on Sharing All Influenza Data (GISAID) on each of the main SARS-CoV-2 variants were collated. These data were aggregated by sample collection date and variant using a computational pipeline available here: https://github.com/globaldothealth/covid19-variants-summary, (*Dasgupta and Kraemer, 2022*, copy archived at swh:1:rev:8adf2f756b182711ad1d0b8707c44d3703786d23). The GISAID data were downloaded on 11 April 2022; Pango lineage designation v1.2.133 was used (*pango-designation, 2022*). We used these data to define calendar time periods when the Omicron variant represented the majority of infections in each country, and also periods during which the Omicron variant represented only a small (<10%) fraction of infections. For each country, the period during which infections were assumed to be caused by other variants ended in the epidemiological week before the Omicron variant relative frequency crossed a low threshold percentage (e.g. 10%) (see *Figure 1*). The first epidemiological

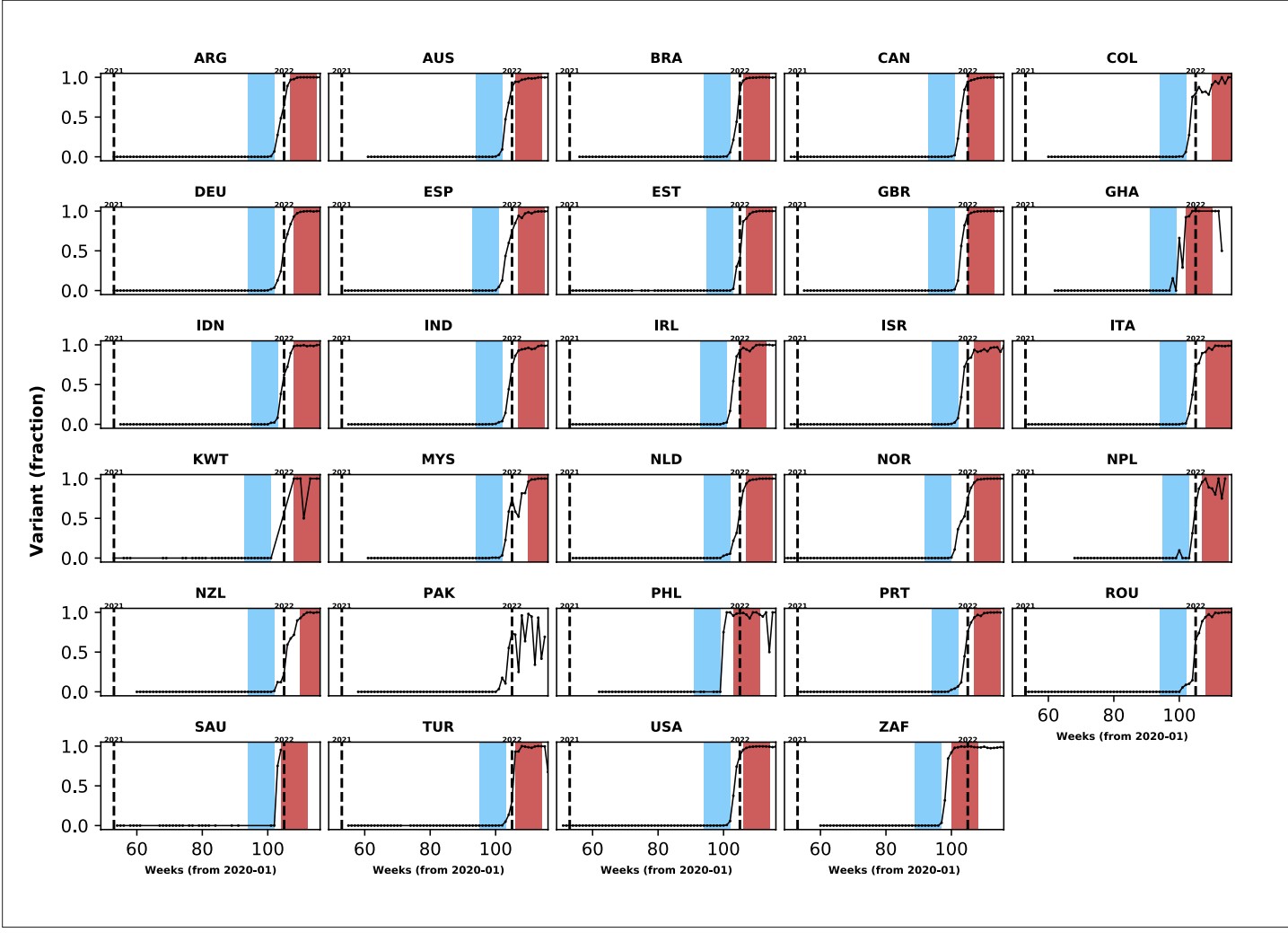

**Figure 1.** Population-level relative frequency of Omicron variant infections by country and time. Here, data aggregated by epidemiological week and country were used to calculate the proportions of infections caused by the Omicron variant. For analyses reported in the *Results* section, two epidemiological periods were defined: the first corresponds to the two months before the Omicron variant reaches a threshold frequency of 10% (blue area in each panel; the *pre-Omicron period*); the second period corresponds to the two months after Omicron variant frequency reaches 90% (red area in each panel; the *Omicron period*). Sensitivity analyses, using other relative frequencies for defining periods, are presented in the Appendix 1. Each panel presents data for a country (ISO3 code as title) contributing clinical data for this analysis; y-axes represent proportions in each epidemiological week (x-axes). Data for Laos are not shown as, at the time of the analysis, samples were not included in the database that informed population-level frequency of Omicron variant during the study period. In Pakistan, due to fluctuations in Omicron variant frequency in the dataset, study periods were not defined. More information on the spread of the Omicron variant in Laos and analysis of the clinical data from Pakistan are presented in the Appendix 1.

week when Omicron variant frequency, as a proportion of all circulating variants, was higher than a given threshold percentage (90% in analyses presented in the *Results* section and 80% in sensitivity analyses) was used as the start date of the period during which all admissions were considered to be caused by the Omicron variant. Note (i) that amongst different countries these two study periods started in different calendar weeks, depending on when the Omicron variant was introduced to the location and on the rate of its local spread, and (ii) that in this analysis all Omicron sub-lineages are included (e.g. BA.1.1, BA.2). Only patients admitted to hospital in the two months before country-level Omicron variant frequency reached the lower threshold and those admitted in the first two months after Omicron variant relative frequency reached 90% were included in the primary analysis; the reason for restricting the study population to those admitted during these time windows was to reduce confounding by unmeasured factors whose frequencies in the hospitalised population also changed over time and which might be associated with clinical outcomes.

## Statistical analysis

We report the frequencies of symptoms, comorbidities and vaccination status stratified by country and time periods (before and after Omicron emergence). We also assessed the case fatality risk and the frequency of a composite outcome that combined death and invasive mechanical ventilation use during the two study periods; in this analysis, patients who were discharged from hospital before the end of the follow-up period used in the definition of the outcome (14 or 28 days) were assumed to have been alive at the end of that period. When estimating risk of death by day 14 after admission or onset of symptoms, whichever happened later, numerators were numbers of patients who died before or on day 14 after admission; denominators in this calculation included those who died by day 14, those discharged at any time during follow-up, and those who were followed at least for 2 weeks, regardless of final outcome, including those who died after 14 days. The same approach was used to analyse the 28 day fatality risk. Note that for 35.5% of patients admitted to hospital during the two study periods defined by Omicron variant frequency, date of onset of symptoms was missing; for these patients we assumed onset of clinical disease happened before admission – that is that these were not hospital acquired infections. Furthermore, for 7.2% of patients, outcome date (date of death or discharge or latest date with follow-up information) was missing and 0.4% had an outcome date that was earlier than date of admission or of symptoms onset; except for those who were discharged and had missing outcome date, these two groups of patients were not included in analyses on the frequencies of clinical outcomes but were included in analyses describing distributions of symptoms and comorbidities. As described in the *Results* section, some patients included in this study were admitted for treatment of a medical condition other than COVID-19 but tested positive incidentally during hospitalisation.

We used mixed-effects logistic regression models to assess the association between study period, that is periods defined by the Omicron variant frequency at the population level, and 14-day death risk, adjusting for age, sex, and vaccination status. Age was included with the following categories: patients younger than 18 years, aged between 18 and 60 years, and older than 60 years. Random intercepts were used to account for potential variation in the risk of death between study sites in different countries. We also present models that adjust for the most commonly reported comorbidities; for each comorbidity included in the analysis, a binary variable was used to indicate presence or absence of the condition. Cox proportional hazards models on time to death, adjusted for age and sex and stratified by country and previous vaccination, were also fit; results of survival analyses are shown in the *Appendix 1*. Note that vaccination status was used as a binary variable in these models, without dose counts or timing of vaccination, and due to limited information on dates of doses we did not adjust for time since the most recent vaccination.

R and Python were used for data processing and descriptive analyses (*R Development Core Team, 2022*; *The pandas development team, 2020*). Code used for analyses and aggregated data used to generate figures are available (*ISARIC Data Platform, 2022*) (see also *Data availability statement*). Stata 17 was used to fit mixed-effects logistic models and perform survival analysis.

## Results

### Description of study population and study periods

Overall, 129,196 records from patients admitted to hospital between 1st October 2021 and 28th February 2022 were included in this analysis. Clinical centres in 30 countries contributed data (median 53 observations per country, interquartile range [IQR] 18–162); 11 countries contributed data on more than 100 hospitalised clinical cases (*Appendix 1—table 1*). A total of 54.0% and 42.6% of records were from South Africa and the United Kingdom, respectively. *Appendix 1—table 2* and *Appendix 1—table 3* show information on missing data for both symptoms and comorbidities.

In addition to the clinical data contributed by the collaborating centres, population-level variant frequency data were used to define time periods when most infections in a country were assumed to be caused by Omicron versus other lineages. As presented in *Figure 1*, different countries reached the threshold relative frequencies of 10 and 90% of infections being caused by the Omicron variant at different times. Similar plots are presented in *Appendix 1—figure 1* for other threshold frequencies. In *Appendix 1—table 4*, we list limitations in the use of these data to define time periods when infections were more likely caused by Omicron versus previous variants.

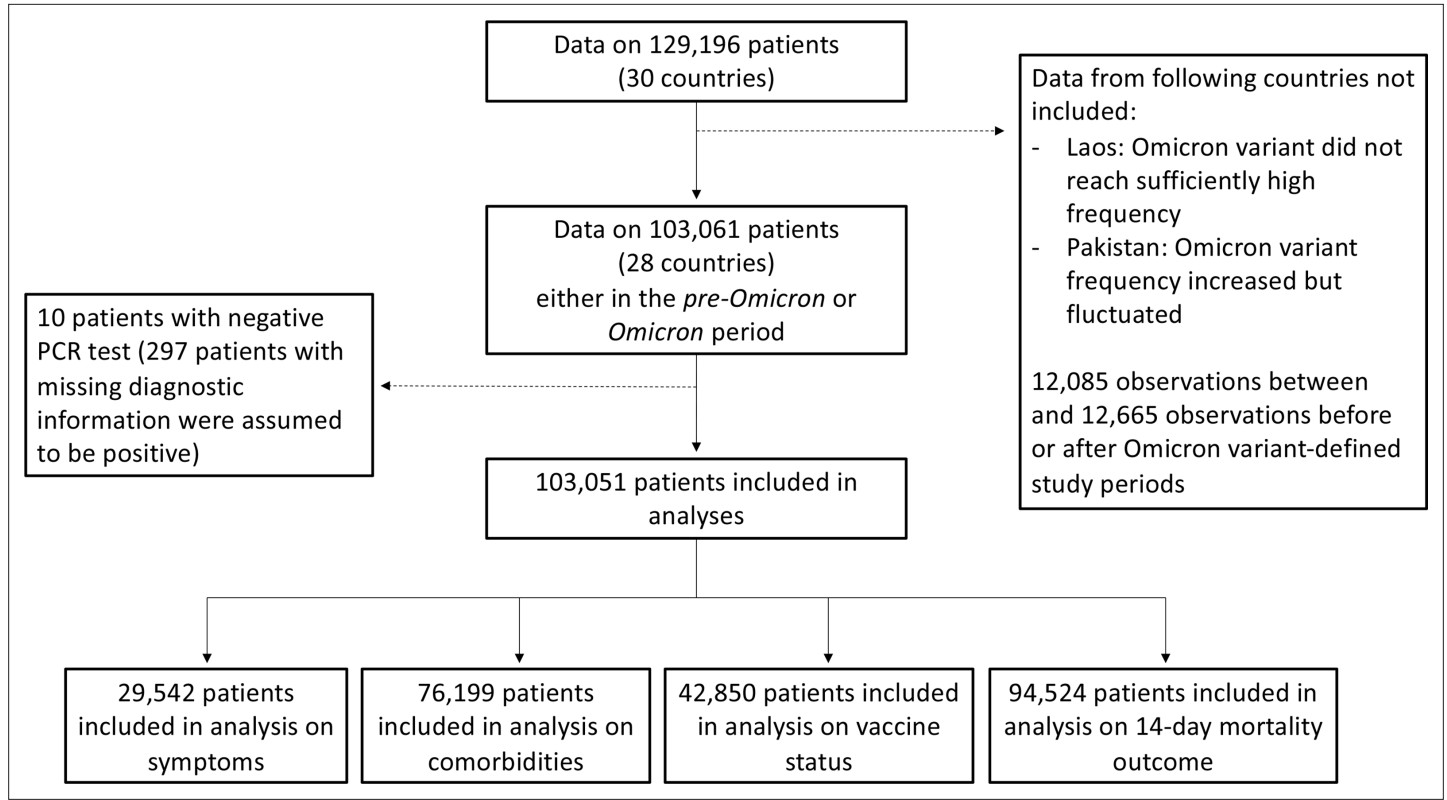

**Figure 2.** Study flowchart. In this figure, we present the numbers of observations included in analyses in the different subsections of the Results section.

Using information presented in *Figure 1*, 103,061 patients, from 28/30 countries, were admitted either in the two months before the Omicron variant represented 10% of infections at the country-level (N=22,921; henceforth, the *pre-Omicron period*) or in the two months after (N=80,140) the Omicron variant was responsible for at least 90% of the infections; for ease of reference, the latter period will be referred to as the *Omicron period*. Note that 12,085 patients were admitted during weeks between the end of the *pre-Omicron period* and the start of the *Omicron period* and are not included in analyses presented in the following subsections (*Figure 2*); and 12,560 records of patients admitted two months after Omicron variant represented 90% of infections were not analysed. All patients from South Africa, the United Kingdom and Malaysia were assumed to be SARS-CoV-2 posi-tive, as this is one criterion for inclusion in their databases. Of the 2296 records from other countries, information on SARS-CoV-2 diagnostic testing was available for 1,999 observations; whilst patients with negative PCR test result (N=10) were excluded from the rest of the analysis, those with missing PCR data (N=297) were assumed positive (see *Appendix 1—table 5* for distribution by country). Of note, clinical data from Laos were not included in comparative analyses as there was only limited evidence of increase in local Omicron variant relative frequency during the study period (additional information is provided in the Appendix 1). For Pakistan, population-level data available at the time of the analysis indicate increasing Omicron variant frequency during the study period, but the propor-tion of local infections caused by this variant fluctuated; analyses of clinical data from that country are described in the Appendix 1.

The median (IQR) ages of patients during the *pre-Omicron* and *Omicron periods* were 62 (43 – 76) and 50 (30 – 72) years, respectively; however, country-specific medians suggest that the younger age of patients after Omicron variant emergence in the combined dataset is at least partially explained by an increase in the proportion of data contributed by South Africa, relative to the proportion of data contributed by other countries (*Appendix 1—table 6*). A total of 48.3% and 54.8% of patients admitted during these periods, respectively, were female. 5.2% and 9.1% of patients in the *pre-Omicron* and *Omicron periods*, respectively, had the date of disease onset after admission date. In some countries, information on whether COVID-19 was the main reason for hospitalisation was also

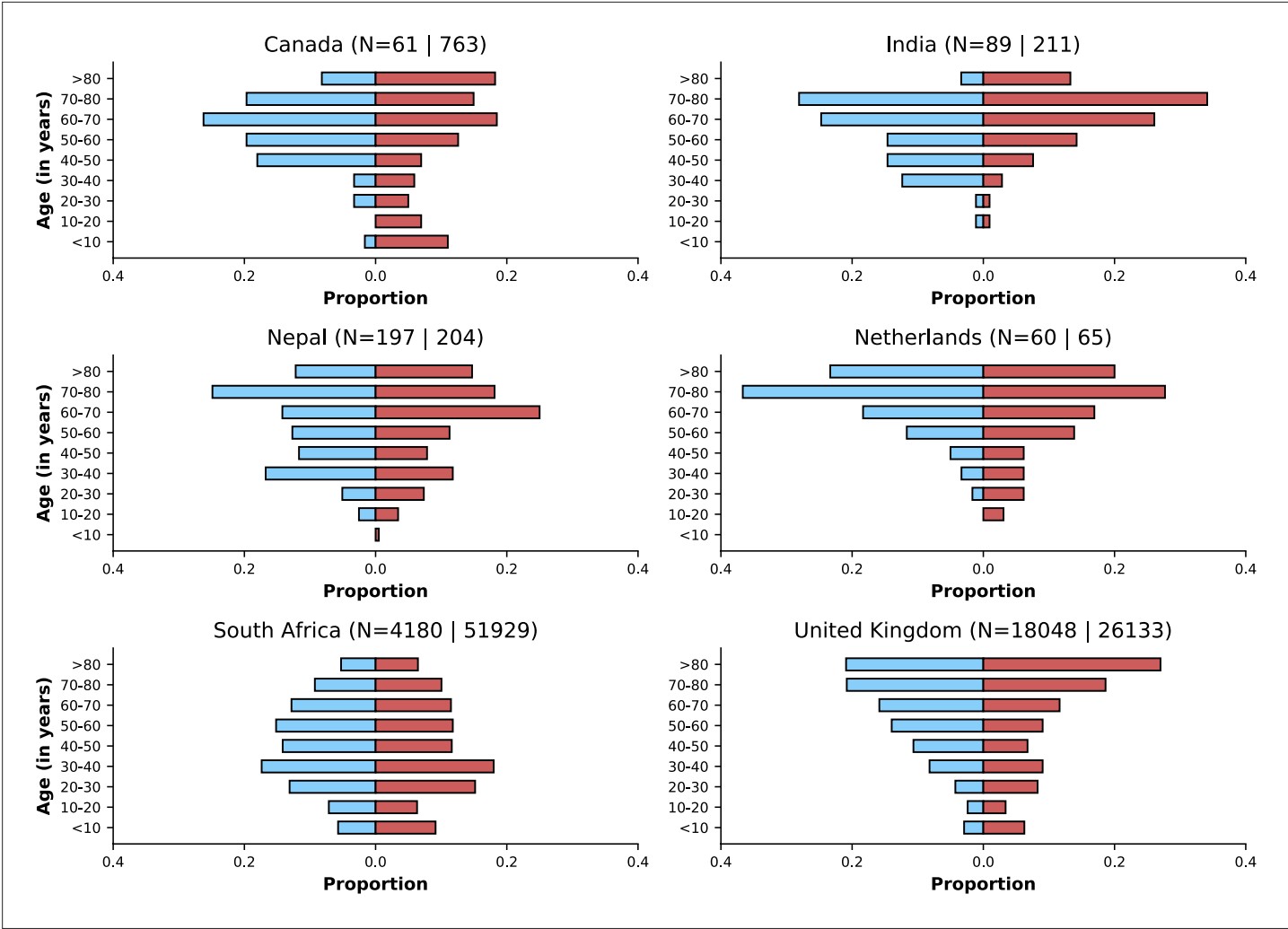

**Figure 3.** Age distributions by study period and country. Age distributions (x-axes show proportions; y-axes, age groups) when Omicron variant relative frequency was below 10% (blue bars) and when the frequency was 90% or higher (red bars). Data from different countries are shown in different panels; only countries with 50 or more records in each period are presented. Numbers of observations with age information are shown for each study period next to country names.

collected: 70.1% (N=2248) and 69.0% (N=27,804) of patients during the *pre-Omicron* and *Omicron periods* respectively were admitted to hospital due to COVID-19; patients for whom this information was available were primarily from South Africa (94.9%). There was no consistent pattern of within-country changes related to this variable (*Appendix 1—table 7*). Of note, 465/36,761 (1.3%) individuals reported a history of previous SARS-CoV-2 infection before the acute episode leading to hospitalisation included in this analysis (128/15,563 [0.8%] and 337/21,198 [1.6%] in the *pre-Omicron* and *Omicron periods*, respectively).

## Temporal changes in frequencies of symptoms and comorbidities

*Figure 3* shows age distributions of hospitalised patients before versus after Omicron variant emergence; only countries with at least 50 observations in each period are included. Despite similar medians of age in the two periods within countries, in some, but not all, country-specific datasets, an increase in the proportion of the study population from younger ages was observed, although the number of patients in some age categories is small. Furthermore, there were differences between countries with regard to age distribution of cases, which could reflect either epidemiological differences between settings or else differences in recruitment of patients for this analysis.

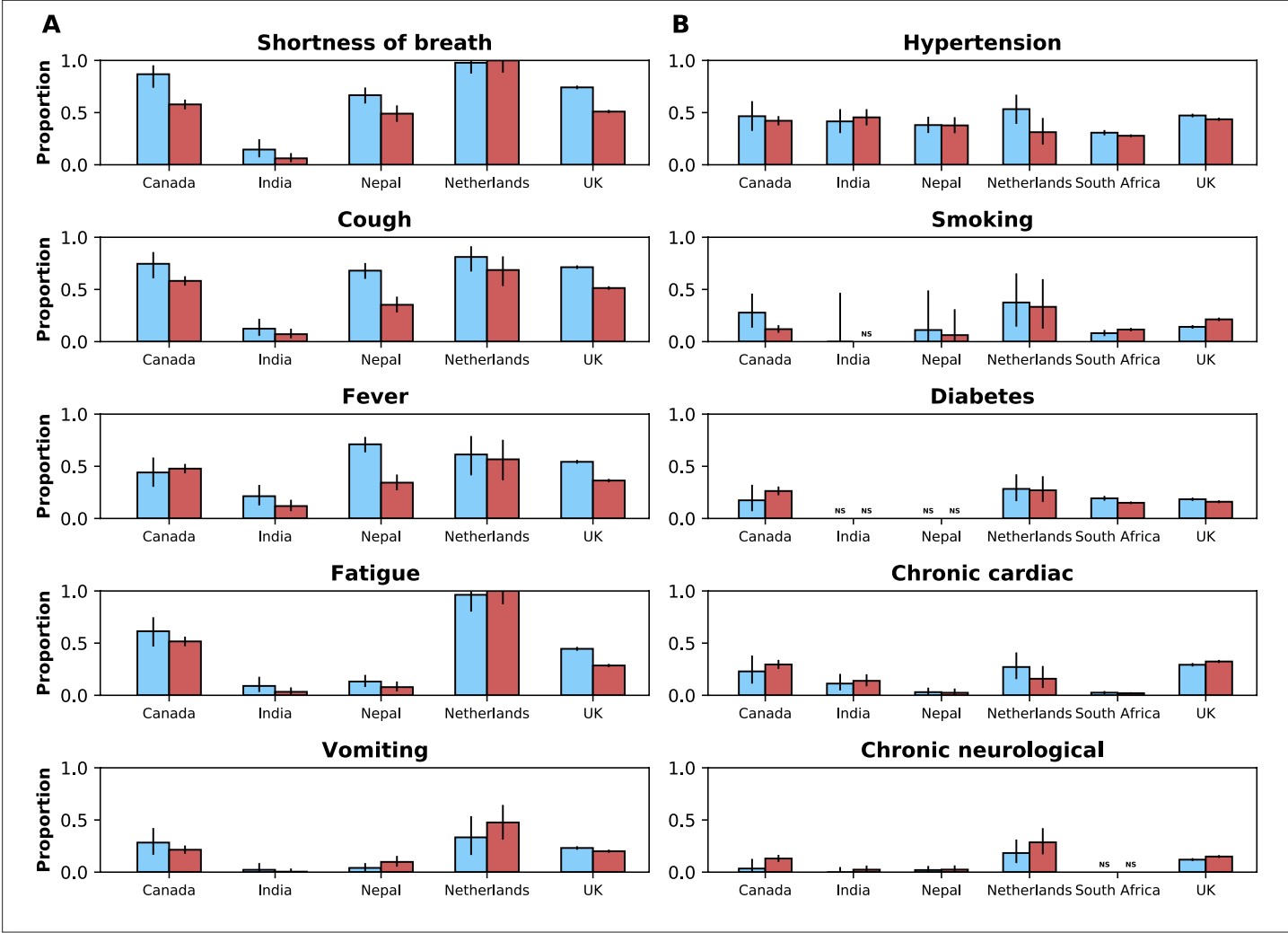

**Figure 4.** Frequencies of symptoms and comorbidities by study period and country. Frequencies of the five most common symptoms (**A**) and comorbidities (**B**) during the *pre-Omicron* (blue bars) and *Omicron* (red bars) *periods*. 95% confidence intervals are shown. Note that South Africa is included in panel B but not panel A. For panel (**A**), only data from the *pre-Omicron period* were used to identify the most frequent symptoms; for panel (**B**), as data on comorbidities were available in the two countries contributing most records, the United Kingdom and South Africa, and since their relative contributions to the study population changed in the two study periods, the dataset including both the *pre-Omicron* and *Omicron periods* was used to identify most common comorbidities. Only countries with at least 50 observations during each study period are included. For each symptom or comorbidity, whenever fewer than five observations without missing data were available, bars were not shown and the text 'NS' (not shown) was included.

The frequencies of the five most commonly reported symptoms and comorbidities in the combined (all countries) dataset during the two study periods are presented in *Figure 4A and B*, by country and study period. When analysing the combined dataset, there was a decrease in the percentage of patients with at least one of the comorbidities listed in *Appendix 1—table 3* before versus during Omicron variant dominance (78.9% [N=15,574] and 59.6% [N=60,625], respectively); however, country-specific data show variable patterns (*Appendix 1—table 8*). With a total of 14 comorbidities being considered, median (IQR) numbers of comorbidity variables with non-missing information in the *pre-Omicron* and *Omicron periods* were 11 (0–12) and 9 (1 – 11), respectively. Whilst the directions of changes (increase or decrease) in frequencies of comorbidities were not consistent across countries, for many symptoms frequencies were lower during the *Omicron period* versus the *pre-Omicron period*. As can be seen in *Appendix 1—figure 2*, this pattern was consistent after stratifying frequencies of symptoms by age groups. The percentage of patients during the *pre-Omicron period* with at least one of the symptoms in *Appendix 1—table 2* was 96.6% (N=11,683); this percentage was 88.6%

**Table 1.** Vaccination status by country and study period.

Data for period-country combinations with less than 10 observations are not presented. Data on vaccination status were not available for patients from Saudi Arabia.

| Country | pre-Omicron period | | Omicron period | |
|---|---|---|---|---|
| | % Vaccinated | Total N | % Vaccinated | Total N |
| Brazil | 84.6 | 13 | 87.9 | 33 |
| Canada | 32.2 | 59 | 57.3 | 686 |
| Colombia | 42.1 | 19 | - | <10 |
| Estonia | - | <10 | | |
| Germany | - | <10 | - | <10 |
| India | 34.8 | 23 | 84.8 | 33 |
| Malaysia | 79.3 | 29 | 80.0 | 10 |
| Nepal | 25.3 | 190 | 39.3 | 183 |
| Netherlands | 60.0 | 60 | 51.0 | 51 |
| New Zealand | 5.9 | 34 | - | <10 |
| Norway | - | <10 | 82.2 | 45 |
| Philippines | 78.6 | 14 | - | <10 |
| Portugal | - | <10 | - | <10 |
| Romania | - | <10 | 78.6 | 98 |
| South Africa | 15.1 | 1605 | 27.9 | 24752 |
| Spain | 45.0 | 20 | 70.9 | 55 |
| United Kingdom | 65.4 | 6865 | 70.3 | 7846 |
| United States of America | - | <10 | - | <10 |
| Argentina | | | - | <10 |
| Australia | | | - | <10 |
| Indonesia | | | - | <10 |
| Israel | | | 54.5 | 11 |
| Kuwait | | | 66.7 | 18 |
| Turkey | | | 74.1 | 27 |

(N=17,859) during the *Omicron period* (see *Appendix 1—table 9* for country-specific numbers). These numbers refer to records from countries other than South Africa, where data on symptoms were not systematically available. The median (IQR) numbers of variables with non-missing data on symptoms were 14 (0–19) and 17 (0–19) for the *pre-Omicron* and *Omicron periods*, respectively.

## Vaccination history in hospitalised patients

Data on vaccination status were available for 42,850/103,051 hospitalised patients (8,952 during the *pre-Omicron period* and 33,898 during the *Omicron period*). In *Table 1*, we present vaccination status for study participants in each of the two periods by country. As expected, there is considerable inter-country variation in the frequency of vaccination. Age-stratified vaccination frequencies are shown in *Appendix 1—figure 3* and suggest increases in frequency of previous vaccination during the period after Omicron variant emergence. However, as shown in *Appendix 1—figure 4*, with population-level vaccination coverage from before Omicron variant emergence up to the end of February 2022, in many countries contributing data to this study there was an increase in vaccination coverage over time, including in the periods during and after the emergence of the Omicron variant. Note that 55.8% of vaccinated patients received two or more doses before hospital admission.

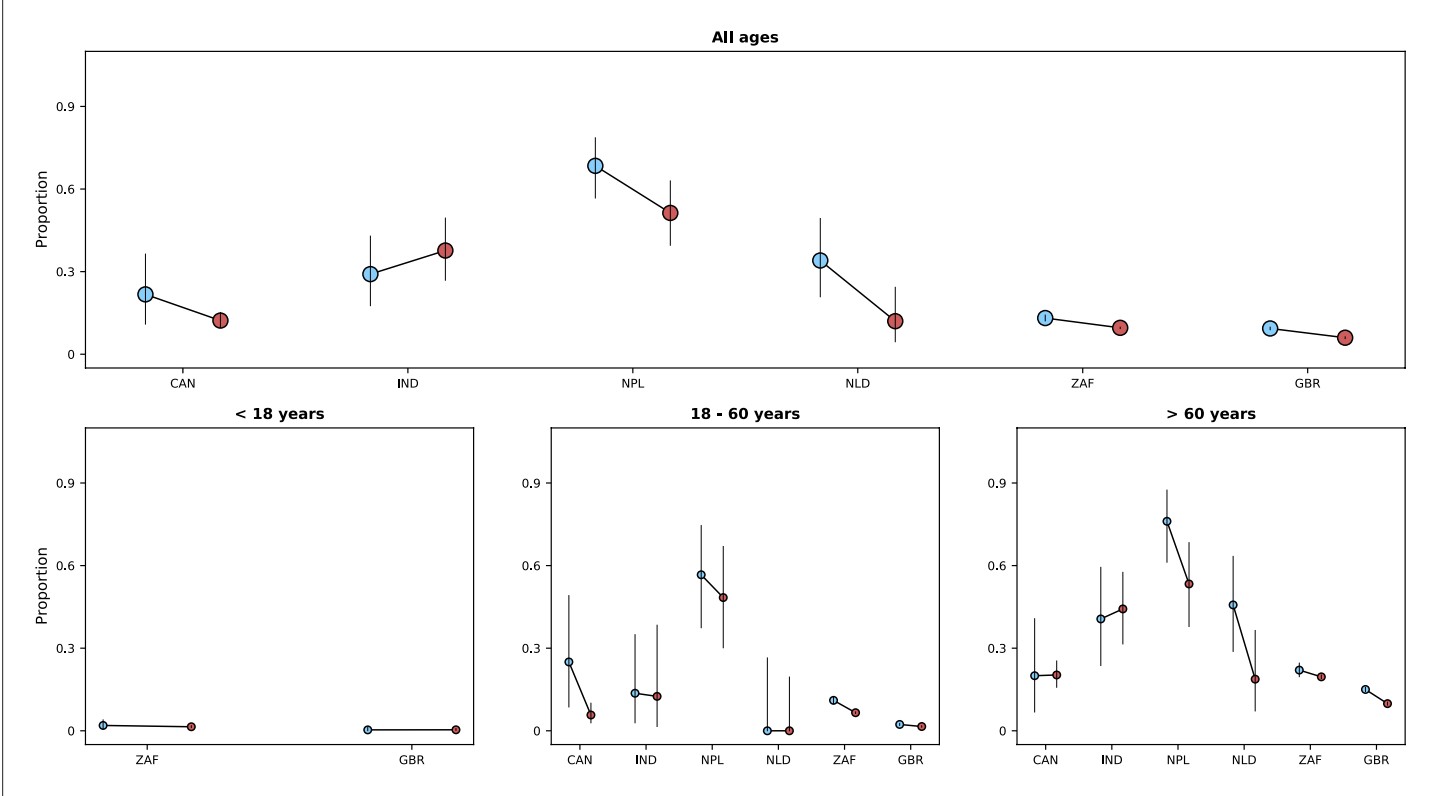

**Figure 5.** Risk of death (y-axes) in the first 14 days after hospital admission or disease onset, whichever occurred latest, during the *pre-Omicron* and *Omicron periods*. In each panel, the x-axis shows countries (ISO3 codes are presented), with different periods represented by circles with different colours (blue circles for the *pre-Omicron period*; red circles, for the *Omicron period*). 95% confidence intervals are also presented. The top panel shows data for individuals of all ages; the bottom panels, data for patients aged less than 18 years, between 18 and 60 years, and older than 60 years. Only countries with at least 50 observations in both study periods are included in the figure; for panels presenting age-specific estimates (bottom row), a further requirement for inclusion was outcome data for at least 10 patients in the corresponding age range in both periods.

## Clinical outcomes

Overall, 11,314 patients admitted during the two study periods died during hospitalisation: 8517/94,524 by day 14 after hospital admission or disease onset, whichever occurred latest, and 10,530/94,461 by day 28; 738 patients died after day 28 and 46 patients who died did not have an outcome date recorded. As explained in the *Methods* section, denominators for fatality risks included patients who were discharged or still in hospital by day 14 or 28. Median (IQR) times to death were 10 (5 – 17) and 6 (3 – 13) days for the periods before and after Omicron emergence, respectively; similar information, on time from admission or symptoms onset to death, stratified by country is shown in *Appendix 1—table 10*. In some countries (see *Figure 5* for comparisons on 14-day fatality risk, and *Appendix 1—figure 5* for comparisons using the 28-day period), during the *Omicron period*, a lower proportion of patients died during hospitalisation, compared to the period before Omicron emergence; in India, the opposite pattern was observed although numbers for that country were limited.

In a mixed-effects logistic model on 14-day fatality risk that adjusted for sex, age categories, and vaccination status, hospitalisations during the *Omicron period* were associated with lower risk of death (see *Table 2*). The inclusion of common comorbidities in the model did not change the estimated association. Similar results were obtained when using 28-day fatality risk as the outcome. We repeated the 14-day fatality risk analysis excluding patients who reported being admitted to hospital due to a medical condition other than COVID-19; the estimated odds ratio for the association between study period and the outcome was similar to those reported in *Table 2*. In an additional sensitivity analysis, estimates from a model that only included data from countries with at least 50 records per study period were also similar (OR 0.65, 95% CI 0.61–0.69, adjusted for covariates included in model I,

**Table 2.** Odds ratio for the association between study period and mortality outcome. Results of multivariate logistic models, with random intercepts for countries, on 14-day fatality risk are presented. Different models were fit that included different variables. Model III adjusts for all variables in the table, however due to missing data in the vaccination and comorbidity variables, less than a third of the study population was included in the estimation of that model; models I and II were thus fit that did not adjust for these variables and included more individuals. In model IV, a category for missing data was created for the variable on previous vaccination; individuals in that category had an odds ratio of 0.74 (0.69–0.80; reference group in this comparison is the non-vaccinated group). Note that similar results were obtained when finer categorisation of the age variable, 10-year intervals, was used. As previous SARS-CoV-2 infection has been shown to reduce severity of COVID-19 (*Altarawneh et al., 2022*), a multivariable model that also adjusted for this variable was fit; in that model, the odds ratio for the association between study period and fatality risk was 0.70 (0.61–0.80). As in other epidemiological studies, estimates for covariates other than the primary exposure (study period) should be carefully interpreted (*Westreich and Greenland, 2013*).

| Model | I | II | III | IV |
|---|---|---|---|---|
| Number of observations | 94,077 | 39,950 | 26,728 | 56,329 |
| | Odds ratio (95% CI) | Odds ratio (95% CI) | Odds ratio (95% CI) | Odds ratio (95% CI) |
| **Variables** | | | | |
| Omicron period* | 0.65 (0.62–0.69) | 0.67 (0.61–0.75) | 0.68 (0.60–0.77) | 0.64 (0.59–0.69) |
| *Sex (male)* | 1.32 (1.26–1.38) | 1.33 (1.23–1.43) | 1.36 (1.24–1.49) | 1.33 (1.25–1.42) |
| *Age* | | | | |
| *Older than 60 years* | Reference | Reference | Reference | Reference |
| *Aged between 18 and 60 years* | 0.26 (0.25–0.27) | 0.24 (0.22–0.26) | 0.27 (0.25–0.30) | 0.30 (0.27–0.32) |
| *Younger than 18 years* | 0.06 (0.05–0.07) | 0.06 (0.05–0.07) | 0.07 (0.05–0.09) | 0.06 (0.05–0.08) |
| *Previous vaccination* | - | 0.60 (0.55–0.65) | 0.53 (0.48–0.59) | 0.59 (0.54–0.65) |
| *Comorbidities* | | | | |
| *Hypertension* | - | - | 1.29 (1.16–1.42) | 1.26 (1.17–1.35) |
| *Diabetes* | - | - | 1.22 (1.09–1.38) | 1.22 (1.12–1.32) |
| *Chronic cardiac disease* | - | - | 1.50 (1.31–1.71) | 1.51 (1.39–1.65) |

*Odds ratio in univariate analysis 0.65 (0.61–0.69) (N=94,524).

*Table 2*). Survival analysis was also performed, and similar results were obtained (*Appendix 1—table 11*).

In addition to using fatality risk in our analyses, we also considered the composite outcome of death or invasive mechanical ventilation (IMV). Data on IMV were available in 74,563 records. Of 74,563 patients, 3111 required IMV during hospitalisation; the date when IMV was initiated was reported for 1070/3111 patients. Of those patients with data on IMV, 10,049/67,383 patients either died or required IMV. *Appendix 1—figure 6* shows proportions of patients with this outcome by country and study period. Since date of IMV initiation was only available for 1070/3111 records, we do not present graphs by time since admission date.

## Comparison with individual-level variant data

Whilst our approach of using population-level variant composition information allowed inclusion in this analysis of data from settings where it was not feasible to systematically identify the infecting SARS-CoV-2 variant, the use of aggregated data to infer the infecting variant has limitations, including the possibility of misclassification (see *Appendix 1—table 4* for a list of limitations of this approach).

To assess whether patterns described in previous subsections are generally consistent with analyses using individual-level variant information, we repeated comparisons for countries where information on the infecting variant was collected; data on variant were available for 1275 records. Of these, 852 patients were admitted either during the *pre-Omicron period* or the *Omicron period*: whilst only 1.9% (16/827) of those admitted during the *Omicron period* were infected by a variant other than Omicron, 4.0% (1/25) of patients during the *pre-Omicron period* had Omicron as the causative virus variant; for the calculation of these percentages data from a participating institution that prioritised contributing Omicron variant cases were not included. Except for six clinical cases in South Africa and Saudi Arabia, all infections were caused either by Delta or Omicron variants, and for this reason only data on these two variants are presented (*Appendix 1—table 12*). Figures similar to *Figures 3–5* but stratified by infecting variant, rather than study period, are shown in the Appendix 1 (*Appendix 1—figures 7–10*). The numbers of participants included in the latter comparisons are lower than the numbers included in the comparisons using population-level variant data; for countries with ten or more observations of both Omicron and Delta variants, the patterns observed are broadly consistent with results obtained using the population-level approach.

We also performed sensitivity analyses using different population-level threshold frequencies for the Omicron variant (10% and 80%, rather than 10% and 90%); these are shown in *Appendix 1—figures 11–14* and are consistent with findings described in the *Results* section.

## Discussion

When new variants of SARS-CoV-2 emerge during the COVID-19 pandemic, several critical questions are asked by public health authorities as to differences in disease severity and risk factors, and vaccine protection. Here, we leveraged data from multiple sources, from population-level variant frequency information to individual-level data on the clinical journey of hospitalised patients with COVID-19, and from multiple countries, to compare characteristics of patients with infection during periods before Omicron emergence versus when this variant became locally dominant. We observed that when the relative frequency of the Omicron variant was high, the proportions of patients with some of the most common COVID-19 symptoms were lower compared to the *pre-Omicron period*. In most but not all countries, patients presenting to hospital during the *Omicron period* had better outcomes (lower fatality risk), compared to those hospitalised before Omicron emergence, which could be related to lower variant virulence, prior immunity or residual confounding. In summary, our approach, which was consistent with analyses that used individual-level variant data from a subset of the study population, suggests clinical differences in patients hospitalised with the Omicron variant versus those admitted before this variant spread, and these differences vary by country.

Our finding that mortality was generally lower during the period when the Omicron variant was dominant is consistent with data from South Africa reported earlier this year (*Wolter et al., 2022*). In that study, which included more than 30,000 patients with individual-level information on the infecting variant, individuals infected with the Omicron variant had a lower risk of disease progression that required hospital admission than individuals infected with other variants; amongst hospitalised patients, the odds ratio for the association between Omicron variant infection and severe disease was 0.7 (95% confidence interval [CI] 0.3–1.4), which is similar to that observed in this study using death as the outcome. A lower risk of death in Omicron variant-infected versus Delta variant-infected patients was also observed in a recent study in the United Kingdom, although that analysis did not assess risk of death conditional on hospitalisation but rather on infection (*Nyberg et al., 2022*). In our analyses, statistical models were adjusted for vaccination history, which is a potential confounder of the association between dominant variant period and risk of death. However, the simplistic approach of using vaccination as a binary variable may be subject to residual confounding by time since vaccination, number of doses, or vaccine type. Moreover, as part of the effort to characterise Omicron variant infection, information on whether COVID-19 was the main reason for hospitalisation was collected during the study period and suggests that for a non-negligible proportion of patients other clinical conditions might have prompted hospital admission. All these factors might have contributed to the observed association, possibly to different degrees in different countries, reason for which this result should not be assumed to necessarily relate to the differences in variant virulence previously suggested by mechanistic studies (*Shuai et al., 2022*; *Halfmann et al., 2022*). Of note, data from India (see *Figure 5*) suggest slightly higher fatality risk during the *Omicron period*

compared to the *pre-Omicron period* for patients older than 60 years, which could be potentially explained by confounding unrelated to age, residual age-related confounding, not controlled by the categorisation used in our analysis, or alternatively by the limited sample size and consequent uncertainty.

During the period of Omicron variant dominance, fewer patients presented with the symptoms most commonly reported earlier. For example, we observed in the United Kingdom that shortness of breath was present in about three-quarters of patients before Omicron variant emergence and in about half of patients during the *Omicron period*. Notably, a similar pattern was observed in Nepal, where patients were more often recruited from critical care settings. One possible explanation for this finding would be if incidental SARS-CoV-2 infections, that is infections that were not the primary reason for hospitalisation, were more frequent during the *Omicron period*; the high transmissibility of this variant, and the consequent peaks in numbers of infections, together with its reported association with lower severity, provides support for this hypothesis. However, in the subset of patients with data on the reason for hospitalisation there was no increase in the proportion of admissions thought to be incidental infections and indeed proportions in both study periods were consistent with frequencies of incidental infections in recent studies in the United States (*Klann et al., 2022*) and the Netherlands (*Voor In 't Holt et al., 2022*), although in the latter, non-incidental infections included patients for whom COVID-19 was a contributing but not the main cause of hospitalisation. An alternative and less plausible explanation for the lower frequency of symptoms during the *Omicron period* would be that some of these patients developed symptoms other than those presented here, and which are severe enough to prompt hospital admission. Finally, it is also possible that the question on the primary reason for hospitalisation might have been interpreted differently in different countries and even in different hospitals in the same country, which would complicate its use in identifying incidental infections.

We also observed that history of COVID-19 vaccination was more frequent during the *Omicron period*, although for most countries the number of patients with vaccination information was limited, especially after stratification by age. Whilst this pattern would be expected if current vaccines were less effective against the Omicron variant compared to previously circulating variants, as suggested by a recent study in England analysing symptomatic disease (*Andrews et al., 2022a*), there were changes in vaccination coverage in many settings during the second half of 2021 and early 2022, including in response to the reports of Omicron variant cases. Since non-COVID-19 patients (*e.g.*, patients with respiratory infections caused by other pathogens) were not systematically recruited for this multi-country study, it is not possible to estimate vaccine effectiveness during the two study periods and assess its change (*Andrews et al., 2022b*).

The major strength of our study relates to inclusion of data from all WHO geographic regions, collected with standardised forms, with over 100,000 records. However, we note that 96.6% of patients were from two countries - South Africa and the United Kingdom - and that the relative contributions of these countries to the study data were different in the two study periods (*Appendix 1—table 5*); to avoid misinterpretations linked to changes in country-specific contributions to data in the *pre-Omicron* and *Omicron periods*, we present descriptive analyses by country and use statistical models that adjust for country-level variation. It is also important to consider the relative contributions of these countries when interpreting descriptive analyses that refer to the combined dataset. Other limitations of our study relate, as mentioned in *Appendix 1—table 4*, to the use of population-level variant data to define periods when infections were likely caused by Omicron variant. For example, if infection by Omicron variant is associated with lower severity and if samples used to inform population-level frequency were often from community cases, then these aggregated data might not represent variant frequency in the hospitalised population. Another weakness of our study is that recruitment procedure was not standardised and was defined locally. Whilst this likely affected the generalisability of our descriptive estimates (fatality risk and frequencies of symptoms and comorbidities) to local populations of hospitalised COVID-19 cases (*Lash et al., 2021*; *Rothman et al., 2013*), it might not have affected the association between study period and fatality risk, at least not beyond the well-described potential for collider bias in hospital-based studies on COVID-19 outcomes (*Griffith et al., 2020*). Finally, missing information on symptoms for patients from South Africa prevented our descriptive analysis of changes in clinical presentation in an African setting. However, despite potential weaknesses in this approach, our results are consistent with reports from South Africa and

elsewhere (*Wolter et al., 2022*), and individual-level variant data available for this study population often matched the two study periods defined by Omicron variant frequency.

In conclusion, we believe our approach of comparing changes in clinical characteristics of COVID-19 using multi-country standardised data, especially when combined with smaller scale studies that collect individual-level data on infecting variants for validation, will be useful in understanding the impact of new variants in the future. Another application will be in using routinely collected health data for cross-country comparisons of variant characteristics. Equally importantly, the successful conduct of this study, and the lessons learned, including the potential weaknesses discussed above, shows that multi-country efforts to study emerging SARS-CoV-2 variants are feasible, improvable and can generate insights to inform policy decision making.

## Acknowledgements

The investigators acknowledge the philanthropic support of the donors to the University of Oxford's COVID-19 Research Response Fund; COVID clinical management team, AIIMS, Rishikesh, India; COVID-19 Clinical Management team, Manipal Hospital Whitefield, Bengaluru, India; Italian Ministry of Health "Fondi Ricerca corrente–L1P6" to IRCCS Ospedale Sacro Cuore–Don Calabria; and Preparedness work conducted by the Short Period Incidence Study of Severe Acute Respiratory Infection.

This work uses data provided by patients and collected by the NHS as part of their care and support #DataSavesLives. The data used for this research were obtained from ISARIC4C. We are extremely grateful to the 2648 frontline NHS clinical and research staff and volunteer medical students who collected these data in challenging circumstances; and the generosity of the patients and their families for their individual contributions in these difficult times. We also acknowledge the support of Jeremy J Farrar and Nahoko Shindo.

## Additional information

### Group author details

**ISARIC Clinical Characterisation Group**

**Sheryl Ann Abdukahil; Audrey Dubot-Pérès; José Antonio Lepe; Ali Abbas; Kamal Abu Jabal; Nashat Abu Salah; Francisca Adewhajah; Enrico Adriano; Marina Aiello; Kate Ainscough; Eman Al Qasim; Angela Alberti; Beatrice Alex; Abdulrahman Al-Fares; Phoebe Ampaw; Sophia Ankrah; Ardiyan Apriyana; Yaseen Arabi; Antonio Arcadipane; Patrick Archambault; Lukas Arenz; Christel Arnold-Day; Ana Aroca; Rakesh Arora; Diptesh Aryal; Elizabeth A Ashley; AM Udara Lakshan Attanyake; Benjamin Bach; J Kenneth Baillie; Valeria Balan; Irene Bandoh; Renata Barbalho; Wendy S Barclay; Michaela Barnikel; Joaquín Baruch; Diego Fernando Bautista Rincon; Abigail Beane; John Beca; Netta Beer; Husna Begum; David Bellemare; Anna Berenguera; Hazel Bergin; Amar Bhatt; Claudia Bianco; Moirangthem Bikram Singh; Felwa Bin Humaid; Jonathan Bitton; Catherine Blier; Lucille Blumberg; Debby Bogaert; Patrizia Bonelli; Dounia Bouhmani; Thipsavanh Bounphiengsy; Latsaniphone Bountthasavong; Bianca Boxma; Kathy Brickell; Aidan Burrell; Ingrid G Bustos; Eder Caceres; Caterina Caminiti; João Camões; Cecilia Canepa; Janice Caoili; Francesca Carlacci; Gayle Carney; Inês Carqueja; François Martin Carrier; Gail Carson; Silvia Castañeda; Nidyanara Castanheira; Roberta Cavalin; Muge Cevik; Bounthavy Chaleunphon; Adrienne Chan; Meera Chand; Alfredo Antonio Chetta; Julian Chica; Danoy Chommanam; Yock Ping Chow; Nathaniel Christy; Barbara Wanjiru Citarella; Sara Clohisey; Perren J Cobb; Cassidy Codan; Marie Connor; Graham S Cooke; Mary Copland; Amanda Corley; Gloria Crowl; Paula Custodio; Ana da Silva Filipe; Andrew Dagens; Peter Daley; Heidi Dalton; Jo Dalton; Nick Daneman; Emmanuelle A Dankwa; Frédérick D'Aragon; Thushan de Silva; Jillian Deacon; William Dechert; Emmanuelle Denis; Santi Dewayanti; Pathik Dhanger; Yael Dishon; Annemarie B Docherty; Arjen M Dondorp; Maria Donnelly; Christl A Donnelly; Chloe Donohue; Peter Doran; Phouvieng Douangdala; James Joshua Douglas; Joanne Downey; Tom Drake; Murray Dryden; Susanne Dudman; Jake Dunning; Lucian Durham; Anne Margarita Dyrhol-Riise; Marco Echeverria-Villalobos; Giorgio Economopoulos; Michael Edelstein; Martina Escher; Mariano Esperatti; Lorinda Essuman; Amna Faheem; Arabella Fahy; Cameron J Fairfield; Laura Feeney; Carlo Ferrari; Sílvia Ferreira; Claudia**

Figueiredo-Mello; Juan Fiorda; Tom Fletcher; Brigid Flynn; Federica Fogliazza; Patricia Fontela; Simon Forsyth; Robert A Fowler; Marianne Fraher; Diego Franch-Llasat; John F Fraser; Christophe Fraser; Ana Freitas Ribeiro; Nora Fuentes; G Argin; Sérgio Gaião; Linda Gail Skeie; Phil Gallagher; Carrol Gamble; Julia Garcia-Diaz; Esteban Garcia-Gallo; Federica Garofalo; Jess Gibson; Michelle Girvan; Geraldine Goco; Joan Gómez-Junyent; Bronner P Gonçalves; Alicia Gonzalez; Patricia Gordon; Margarite Grable; Christopher A Green; William Greenhalf; Fiona Griffiths; Anja Grosse Lordemann; Anne-Marie Guerguerian; Daniel Haber; Hannah Habraken; Matthew Hall; Sophie Halpin; Summer Hamza; Rashan Haniffa; Hayley Hardwick; Ewen M Harrison; Janet Harrison; Alan Hartman; Madiha Hashmi; Leanne Hays; Lars Hegelund; Lars Heggelund; Ross Hendry; Liv Hesstvedt; Astarini Hidayah; Rupert Higgins; Samuel Hinton; Antonia Ho; Jan Cato Holter; Peter Horby; Juan Pablo Horcajada; Abby Hurd; Samreen Ijaz; Clare Jackson; Nina Jamieson; Waasila Jassat; Synne Jenum; Philippe Jouvet; Dafsah Juzar; Chris Kandel; Christiana Kartsonaki; Anant Kataria; Kevin Katz; Hannah Keane; Seán Keating; Yvelynne Kelly; Sadie Kelly; Kalynn Kennon; Sharma Keshav; Imrana Khalid; Michelle E Kho; Saye Khoo; Peter Kiiza; Beathe Kiland Granerud; Anders Benjamin Kildal; Anders Kildal; Paul Klenerman; Gry Kloumann Bekken; Stephen R Knight; Robin Kobbe; Paa Kobina Forson; Chamira Kodippily; Franklina Korkor Abebrese; Volkan Korten; Karolina Krawczyk; Deepali Kumar; Demetrios Kutsogiannis; Ama Kwakyewaa Bedu-Addo; François Lamontagne; Marina Lanza; Nicola Latronico; Andy Law; Teresa Lawrence; James Lee; Jennifer Lee; Gary Leeming; Amy Lester-Grant; Andrew Letizia; Gianluigi Li Bassi; Janet Liang; Wei Shen Lim; Andreas Lind; Ruth Lyons; Giuseppe Maglietta; Maria Majori; Paddy Mallon; Patrizia Mammi; Frank Manetta; Ceila Maria Sant Ana Malaque; Daniel Marino; Carlos Cañada Illana; Catherine Marquis; Hannah Marrinan; Laura Marsh; John Marshall; Dori-Ann Martin; Ignacio Martin-Loeches; Alejandro Martin-Quiros; Alejandro Martín-Quiros; Caroline Martins Rego; Gennaro Martucci; Eva Miranda Marwali; David Maslove; Sabina Mason; Henrique Mateus Fernandes; Romans Matulevics; Mayfong Mayxay; Colin McArthur; Anne McCarthy; Rachael McConnochie; Sarah E McDonald; Allison McGeer; Johnny McKeown; Kenneth A McLean; Elaine McPartlan; Edel Meaney; Kusum Menon; Alexander J Mentzer; Laura Merson; Tiziana Meschi; Dan Meyer; Alison M Meynert; Efstathia Mihelis; Elena Molinos; Brenda Molloy; Claudia Montes; Shona C Moore; Sarah Moore; Lina Morales Cely; Caroline Mudara; Fredrik Müller; Karl Erik Müller; Laveena Munshi; Lorna Murphy; Srinivas Murthy; Himed Musaab; Carlotta Mutti; Himasha Muvindi; Mangala Narasimhan; Matthew Nelder; Emily Neumann; Alistair Nichol; Lisa Norman; Mahdad Noursadeghi; Giovanna Occhipinti; Derbrenn OConnor; Katie O'Hearn; Piero L Olliaro; David SY Ong; Wilna Oosthuyzen; Peter Openshaw; Linda O'Shea; Massimo Palmarini; Giovanna Panarello; Prasan Kumar Panda; Hem Paneru; Paolo Parducci; Rachael Parke; Melissa Parker; Laura Patrizi; Lisa Patterson; Mical Paul; William A Paxton; Mare Pejkovska; Luis Periel; Michele Petrovic; Frank Olav Pettersen; Scott Pharand; Ooyanong Phonemixay; Soulichanya Phoutthavong; Roberta Pisi; Riinu Pius; Simone Piva; Georgios Pollakis; Andra-Maris Post; Jeff Powis; Viladeth Praphasiri; Mark G Pritchard; Gamage Dona Dilanthi Priyadarshani; Matteo Puntoni; Vilmaris Quinones-Cardona; Else Quist-Paulsen; Anais Rampello; Rajavardhan Rangappa; Elena Ranza; Aasiyah Rashan; Thalha Rashan; Indrek Rätsep; Cornelius Rau; Francesco Rausa; Brenda Reeve; Liadain Reid; Dag Henrik Reikvam; Jordi Rello; Oleksa Rewa; Luis Felipe Reyes; Asgar Rishu; Maria Angelica Rivera Nuñez; Stephanie Roberts; David L Robertson; Ferran Roche-Campo; Amanda Rojek; Roberto Roncon-Albuquerque; Matteo Rossetti; Sandra Rossi; Clark D Russell; Aleksander Rygh Holten; Luca Sacchelli; Musharaf Sadat; Valla Sahraei; Leonardo Salazar; Kizy Sanchez de Oliveira; Vanessa Sancho-Shimizu; Gyan Sandhu; Zulfiqar Sandhu; Oana Sandulescu; Marlene Santos; Shirley Sarfo-Mensah; Iam Claire E Sarmiento; Sree Satyapriya; Rumaisah Satyawati; Egle Saviciute; Gary Schwartz; Janet T Scott; James Scott-Brown; Malcolm G Semple; Ellen Shadowitz; Shaikh Sharjeel; Catherine A Shaw; Victoria Shaw; Rajesh Mohan Shetty; Haixia Shi; Mohiuddin Shiekh; Sally Shrapnel; Moses Siaw-Frimpong; Bountoy Sibounheuang; Louise Sigfrid; Piret Sillaots; Budha Charan Singh; Pompini Agustina Sitompul; Vegard Skogen; Sue Smith; Michelle Smyth; Tom Solomon; Rima Song; BP Sanka Ruwan Sri Darshana; Shiranee Sriskandan; Stephanie-Susanne Stecher; Trude Steinsvik; Birgitte Stiksrud; Adrian Streinu-Cercel; Anca Streinu-Cercel; David Stuart; Jacky Y Suen; Charlotte Summers; Jaques Sztajnbok; Maria Lawrensia Tampubolon; Richard S Tedder; Hubert Tessier-Grenier; Shaun Thompson; David Thomson; Emma C Thomson; Ryan S Thwaites; Andrea Ticinesi; Paul Tierney; Tirupakuzhi Vijayaraghavan; Kristian Tonby; Rosario Maria Torres

Santos-Olmo; Lance CW Turtle; Anders Tveita; PG Ishara Udayanga; Alberto Uribe; Timothy M Uyeki; Ilaria Valzano; Pooja Varghese; Michael Varrone; Sebastian Vencken; James Vickers; José Ernesto Vidal; Judit Villar; Andrea Villoldo; Chiara Vitiello; Manivanh Vongsouvath; Steve Webb; Jia Wei; Sanne Wesselius; Murray Wham; Nicole White; Surya Otto Wijaya; Evert-Jan Wils; Xin Ci Wong; Stephanie Yerkovich; Touxiong Yiaye; Obada Yousif; Saptadi Yuliarto; Maram Zahran; Maria Zambon

## Competing interests

Srinivas Murthy: declares receiving salary support from the Health Research Foundation and Innovative Medicines Canada Chair in Pandemic Preparedness Research. Malcolm G Semple: reports grants from DHSC National Institute of Health Research UK, from the Medical Research Council UK, and from the Health Protection Research Unit in Emerging & Zoonotic Infections, University of Liverpool, supporting the conduct of the study; other interest in Integrum Scientific LLC, Greensboro, NC, USA, outside the submitted work. Alejandro Martín-Quiros: declares consulting fees for Gilead and MSD, presentation fees for GILEAD, Pfizer and MSD, support for attending ECCMID from Gilead, and advisory board fees for MSD and Gilead. Andrea Angheben: declares support from Italian Ministry of Health - "Fondi Ricerca Corrente" Line1 Project 5 to IRCCS Sacro Cuore - Don Calabria Hospital. François Martin Carrier: declares a grant from the Canadian Institute of Health Research. Joan Gómez-Junyent: declares support by Pfizer, Angelini and MSD to attend meetings (registration to meetings only). James Joshua Douglas: declares personal fees from lectures from Sunovion and Merck and consulting fees from Pfizer. Anca Streinu-Cercel: has been an investigator in COVID-19 clinical trials by Algernon Pharmaceuticals, Atea Pharmaceuticals, Regeneron Pharmaceuticals, Diffusion Pharmaceuticals, Celltrion, Inc and Atriva Therapeutics, outside the scope of the submitted work. Rob Fowler: declares a peer reviewed research grant from the Canadian Institutes of Health Research. ISARIC Clinical Characterisation Group: See Appendix 1 for competing interests for group author members. The other authors declare that no competing interests exist.

## Funding

| Funder | Grant reference number | Author |
| --- | --- | --- |
| UK Foreign, Commonwealth and Development Office | | Bronner P Gonçalves<br>Peter Horby<br>Gail Carson<br>Piero L Olliaro<br>Valeria Balan<br>Barbara Wanjiru Citarella<br>Janice Caoili<br>Madiha Hashmi |
| Wellcome Trust | 215091/Z/18/Z | Bronner P Gonçalves<br>Peter Horby<br>Gail Carson<br>Piero L Olliaro<br>Valeria Balan<br>Barbara Wanjiru Citarella |
| Wellcome Trust | 222410/Z/21/Z | Bronner P Gonçalves<br>Peter Horby<br>Gail Carson<br>Piero L Olliaro<br>Valeria Balan<br>Barbara Wanjiru Citarella |
| Wellcome Trust | 225288/Z/22/Z | Bronner P Gonçalves<br>Peter Horby<br>Gail Carson<br>Piero L Olliaro<br>Valeria Balan<br>Barbara Wanjiru Citarella |
| Wellcome Trust | 222048/Z/20/Z | Janice Caoili<br>Madiha Hashmi |

| Funder | Grant reference number | Author |
|---|---|---|
| Bill and Melinda Gates Foundation | OPP1209135 | Peter Horby<br>Gail Carson<br>Piero L Olliaro<br>Joaquín Baruch |
| University of Oxford's COVID-19 Research Fund | 0009146 | Laura Merson |
| Branco Weiss Fellowship | | Moritz UG Kraemer |
| Google.org | | Moritz UG Kraemer |
| Oxford Martin School, University of Oxford | | Moritz UG Kraemer |
| Rockefeller Foundation | | Moritz UG Kraemer |
| European Union Horizon 2020 | 874850 | Moritz UG Kraemer |
| CIHR Coronavirus Rapid Research Funding Opportunity | OV2170359 | François Martin Carrier<br>Srinivas Murthy<br>Asgar Rishu<br>Rob Fowler<br>James Joshua Douglas |
| Bevordering Onderzoek Franciscus | | David SY Ong |
| Italian Ministry of Health "Fondi Ricerca corrente-L1P6" | | Andrea Angheben |
| National Institute for Health Research | CO-CIN-01 | J Kenneth Baillie<br>Malcolm G Semple<br>Ewen M Harrison |
| Medical Research Council | MC_PC_19059 | J Kenneth Baillie<br>Malcolm G Semple<br>Ewen M Harrison |
| NIHR Health Protection Research Unit (HPRU) in Emerging and Zoonotic Infections | 200907 | J Kenneth Baillie<br>Malcolm G Semple<br>Ewen M Harrison |
| NIHR HRPU in Respiratory Infections | 200927 | J Kenneth Baillie<br>Malcolm G Semple<br>Ewen M Harrison |
| Liverpool Experimental Cancer Medicine Centre | C18616/A25153 | Malcolm G Semple |
| NIHR Biomedical Research Centre | IS-BRC-1215-20013 | J Kenneth Baillie<br>Malcolm G Semple<br>Ewen M Harrison |
| NIHR Clinical Research Network | | J Kenneth Baillie<br>Malcolm G Semple<br>Ewen M Harrison |

The funders had no role in study design, data collection and interpretation, or the decision to submit the work for publication. For the purpose of Open Access, the authors have applied a CC BY public copyright license to any Author Accepted Manuscript version arising from this submission.

## Author contributions

Bronner P Gonçalves, Conceptualization, Software, Formal analysis, Investigation, Visualization, Methodology, Writing – original draft, Writing – review and editing; Matthew Hall, Conceptualization, Data curation, Software, Investigation, Methodology, Writing – review and editing; Waasila Jassat, Srinivas Murthy, Malcolm G Semple, Yaseen Arabi, Madiha Hashmi, Gail Carson, Conceptualization, Investigation, Writing – review and editing; Valeria Balan, Data curation, Investigation, Project administration,

Writing – review and editing; Christiana Kartsonaki, Rob Fowler, Conceptualization, Investigation, Methodology, Writing – review and editing; Amanda Rojek, Joaquín Baruch, Conceptualization, Investigation, Methodology, Writing – original draft, Writing – review and editing; Luis Felipe Reyes, Janet Diaz, Investigation, Methodology, Writing – review and editing; Abhishek Dasgupta, Data curation, Software, Investigation, Methodology, Writing – review and editing; Jake Dunning, Mark Pritchard, Conceptualization, Investigation, Methodology; Barbara Wanjiru Citarella, Data curation, Investigation, Methodology, Project administration, Writing – review and editing; Alejandro Martín-Quiros, Andrea Angheben, David SY Ong, Evert-Jan Wils, Investigation, Writing – review and editing; Uluhan Sili, J Kenneth Baillie, Diptesh Aryal, Aasiyah Rashan, Janice Caoili, François Martin Carrier, Ewen M Harrison, Joan Gómez-Junyent, Claudia Figueiredo-Mello, James Joshua Douglas, Mohd Basri Mat Nor, Yock Ping Chow, Xin Ci Wong, Silvia Bertagnolio, Soe Soe Thwin, Anca Streinu-Cercel, Leonardo Salazar, Asgar Rishu, Rajavardhan Rangappa, Investigation; Moritz UG Kraemer, Data curation, Investigation, Methodology, Writing – review and editing; Peter Horby, Piero L Olliaro, Conceptualization, Funding acquisition, Investigation, Methodology, Writing – original draft; Laura Merson, Conceptualization, Funding acquisition, Investigation, Writing – original draft; ISARIC Clinical Characterisation Group, Funding acquisition

## Author ORCIDs
Bronner P Gonçalves http://orcid.org/0000-0002-3329-6050
Matthew Hall http://orcid.org/0000-0002-2671-3864
Abhishek Dasgupta http://orcid.org/0000-0003-4420-0656
Uluhan Sili http://orcid.org/0000-0002-9939-9298
J Kenneth Baillie http://orcid.org/0000-0001-5258-793X
Mohd Basri Mat Nor http://orcid.org/0000-0002-5433-6357
Xin Ci Wong http://orcid.org/0000-0002-1036-8023
Evert-Jan Wils http://orcid.org/0000-0002-2868-0920
Laura Merson http://orcid.org/0000-0002-4168-1960

## Ethics

Ethics Committee approval for this work was given by the World Health Organisation Ethics Review Committee (RPC571 and RPC572 on 25 April 2013). Institutional approval was additionally obtained by participating sites including the South Central Oxford C Research Ethics Committee in England (Ref 13/SC/0149) and the Scotland A Research Ethics Committee (Ref 20/SS/0028) for the United Kingdom and the Human Research Ethics Committee (Medical) at the University of the Witwatersrand in South Africa as part of a national surveillance programme (M160667) collectively representing the majority of the data. Other institutional and national approvals are in place as per local requirements.

## Decision letter and Author response
Decision letter https://doi.org/10.7554/eLife.80556.sa1
Author response https://doi.org/10.7554/eLife.80556.sa2

# Additional files

## Supplementary files
• MDAR checklist

## Data availability

The data that underpin this analysis are highly detailed clinical data on individuals hospitalised with COVID-19. Due to the sensitive nature of these data and the associated privacy concerns, they are available via a governed data access mechanism following review of a data access committee. Data can be requested via the IDDO COVID-19 Data Sharing Platform (http://www.iddo.org/covid-19). The Data Access Application, Terms of Access and details of the Data Access Committee are available on the website. Briefly, the requirements for access are a request from a qualified researcher working with a legal entity who have a health and/or research remit; a scientifically valid reason for data access which adheres to appropriate ethical principles. The full terms are at https://www.iddo.org/document/covid-19-data-access-guidelines. A small subset of sites who contributed data to this analysis have not agreed to pooled data sharing as above. In the case of requiring access to these data, please contact

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

## Appendix 1

### Acknowledgements (partner institutions)

The contribution of those listed in the ISARIC Clinical Characterisation Group was supported by a grant from the Oxford University COVID-19 Research Response fund (grant 0009109); endorsement of the Irish Critical Care- Clinical Trials Group, co-ordinated in Ireland by the Irish Critical Care- Clinical Trials Network at University College Dublin and funded by the Health Research Board of Ireland [CTN-2014–12]; grants from Rapid European COVID-19 Emergency Response research (RECOVER) [H2020 project 101003589] and European Clinical Research Alliance on Infectious Diseases (ECRAID) [965313]; Wellcome Trust [Turtle, Lance-fellowship 205228/Z/16/Z]; Research Council of Norway grant no 312780, and a philanthropic donation from Vivaldi Invest A/S owned by Jon Stephenson von Tetzchner; PJMO is supported by the UK's National Institute for Health Research (NIHR) via Imperial's Biomedical Research Centre (NIHR Imperial BRC), Imperial's Health Protection Research Unit in Respiratory Infections (NIHR HPRU RI), the Comprehensive Local Research Networks (CLRNs) and is an NIHR Senior Investigator (NIHR201385); Cambridge NIHR Biomedical Research Centre; Institute for Clinical Research (ICR), National Institutes of Health (NIH) supported by the Ministry of Health Malaysia; Gender Equity Strategic Fund at University of Queensland, Artificial Intelligence for Pandemics (A14PAN) at University of Queensland, The Australian Research Council Centre of Excellence for Engineered Quantum Systems (EQUS, CE170100009), The Prince Charles Hospital Foundation, Australia; the South Eastern Norway Health Authority and the Research Council of Norway; the U.S. DoD Armed Forces Health Surveillance Division, Global Emerging Infectious Diseases Branch to the U.S Naval Medical Research Unit No. TWO (NAMRU-2) (Work Unit #: P0153_21_N2). These authors would like to thank Vysnova Partners, Inc for the management of this research project. The Lao-Oxford-Mahosot Hospital-Wellcome Trust Research Unit is funded by the Wellcome Trust.

The investigators acknowledge the philanthropic support of the COVID clinical management team, AIIMS, Rishikesh, India; COVID-19 Clinical Management team, Manipal Hospital Whitefield, Bengaluru, India; the dedication and hard work of the Groote Schuur Hospital Covid ICU Team, supported by the Groote Schuur nursing and University of Cape Town registrar bodies coordinated by the Division of Critical Care at the University of Cape Town; the dedication and hard work of the Norwegian SARS-CoV-2 study team; and Preparedness work conducted by the Short Period Incidence Study of Severe Acute Respiratory Infection.

This work uses data provided by patients and collected by the NHS as part of their care and support #DataSavesLives. The data used for this research were obtained from ISARIC4C. The authors are extremely grateful to the 2648 frontline NHS clinical and research staff and volunteer medical students who collected these data in challenging circumstances; and the generosity of the patients and their families for their individual contributions in these difficult times. The COVID-19 Clinical Information Network (CO-CIN) data was collated by ISARIC4C Investigators. The authors also acknowledge the support of Jeremy J Farrar and Nahoko Shindo.

### Conflict of interest declarations for the ISARIC Clinical Characterisation Group

A Angheben declares support from Italian Ministry of Health - "Fondi Ricerca Corrente" Line1 Project 5 to IRCCS Sacro Cuore – Don Calabria Hospital.

FM Carrier declares a grant from the Canadian Institute of Health Research.

H Dalton declares personal fees for medical director of Innovative ECMO Concepts and honorarium from Abiomed/BREETHE Oxi-1 and Instrumentation Labs. Consultant fee, Entegrion Inc, Medtronic and Hemocue.

AM Dyrhol-Riise declares grants from Gilead outside this work.

CA Donnelly declares research funding from the UK Medical Research Council and the UK National Institute for Health Research.

JJ Douglas declares personal fees from lectures from Sunovion and Merck; consulting fees from Pfizer.

R Fowler declares a peer reviewed research grant from the Canadian Institutes of Health Research.

J Gómez-Junyent declares support by Pfizer, Angelini and MSD to attend meetings (registration to meetings only).

AM Guerguerian participated as site investigator for the Hospital For Sick Children, Toronto, Canada as a site through SPRINT-SARI Study via the Canadian Critical Care Trials Group sponsored in part by the Canadian Institutes of Health Research.

A Ho declares grant funding from Medical Research Council UK, Scottish Funding Council - Grand Challenges Research Fund, and the Wellcome Trust, outside this submitted work.

JC Holter reports grants from Research Council of Norway grant no 312780, and from Vivaldi Invest A/S owned by Jon Stephenson von Tetzchner, during the conduct of the study.

Kumar, D. declares grants and personal fees from Roche, GSK and Merck; and personal fees from Pfizer and Sanofi.

DJ Kutsogiannis declares personal fees for a lecture from Tabuk Pharmaceuticals and the Saudi Critical Care Society

J Lee reports grants from European Commission PREPARE grant agreement No 602525, European Commission RECOVER Grant Agreement No 101003589 and European Commission ECRAID Grant Agreement 965313 supporting the conduct, coordination and management of the work.

WS Lim declares his institution has received unrestricted investigator-initiated research funding from Pfizer for an unrelated multicentre cohort study in which he is the Chief Investigator, and research funding from the National Institute for Health Research, UK for various clinical trials outside the submitted work.

I Martin-Loeches declared lectures for Gilead, Thermofisher, MSD; advisory board participation for Fresenius Kabi, Advanz Pharma, Gilead, Accelerate, Merck; and consulting fees for Gilead outside of the submitted work.

A Martín-Quiros declares consulting fees for Gilead and MSD, presentation fees for GILEAD, Pfizer and MSD, support for attending ECCMID from Gilead, and advisory board fees for MSD and Gilead.

S Murthy declares receiving salary support from the Health Research Foundation and Innovative Medicines Canada Chair in Pandemic Preparedness Research.

A Nichol declares a grant from the Health Research Board of Ireland to support data collection in Ireland (CTN-2014–012), an unrestricted grant from BAXTER for the TAME trial kidney substudy and consultancy fees paid to his institution from AM-PHARMA.

P Openshaw has served on scientific advisory boards for Janssen/J&J, Oxford Immunotech Ltd, GSK, Nestle and Pfizer (fees to Imperial College). He is Imperial College lead investigator on EMINENT, a consortium funded by the MRC and GSK. He is a member of the RSV Consortium in Europe (RESCEU) and Inno4Vac, Innovative Medicines Initiatives (IMI) from the European Union.

R Parke declares that the Cardiothoracic and Vascular Intensive Care Unit, Auckland City Hospital, receives support by way of an unrestricted grant from Fisher and Paykel Healthcare New Zealand Ltd.

O Rewa declares honoraria from Baxter Healthcare Inc and Leading Biosciences Inc

O Săndulescu has been an investigator in COVID-19 clinical trials by Algernon Pharmaceuticals, Atea Pharmaceuticals, Regeneron Pharmaceuticals, Diffusion Pharmaceuticals, Celltrion, Inc and Atriva Therapeutics, outside the scope of the submitted work.

MG Semple reports grants from DHSC National Institute of Health Research UK, from the Medical Research Council UK, and from the Health Protection Research Unit in Emerging & Zoonotic Infections, University of Liverpool, supporting the conduct of the study; other interest in Integrum Scientific LLC, Greensboro, NC, USA, outside the submitted work.

S Shrapnel participated as an investigator for an observational study analysing ICU patients with COVID-19 (for the Critical Care Consortium including ECMOCARD) funded by The Prince Charles Hospital Foundation during the conduct of this study.

Adrian Streinu-Cercel has been an investigator in COVID-19 clinical trials by Algernon Pharmaceuticals, Atea Pharmaceuticals, Regeneron Pharmaceuticals, Diffusion Pharmaceuticals, and Celltrion, Inc, outside the scope of the submitted work.

Anca Streinu-Cercel has been an investigator in COVID-19 clinical trials by Algernon Pharmaceuticals, Atea Pharmaceuticals, Regeneron Pharmaceuticals, Diffusion Pharmaceuticals, Celltrion, Inc and Atriva Therapeutics, outside the scope of the submitted work.

C Summers reports that she has received fees for consultancy for Abbvie and Roche relating to COVID-19 therapeutics. She was also the UK Chief Investigator of a GlaxoSmithKline plc sponsored

study of a therapy for COVID, and is a member of the UK COVID Therapeutic Advisory Panel (UK-CTAP). Outside the scope of this work, Dr Summers' institution receives research grants from the Wellcome Trust, UKRI/MRC, National Institute for Health Research (NIHR), GlaxoSmithKline and AstraZeneca to support research in her laboratory.

S Dudman reports grants from Research Council of Norway grant no 312780.

R Tedder reports grants from MRC/UKRI during the conduct of the study. In addition, R Tedder has a patent United Kingdom Patent Application No. 2014047.1 "SARS-CoV-2 antibody detection assay" issued.

L Turtle reports grants from MRC/UKRI during the conduct of the study and fees from Eisai for delivering a lecture related to COVID-19 and cancer, paid to the University of Liverpool.

## Supplementary results

### Frequency of symptoms outside the United Kingdom and South Africa

Most, 82.5% (N=579), patients admitted to hospital during the *pre-Omicron period* outside the United Kingdom and South Africa had at least one symptom; this percentage is lower than the frequency estimated including the United Kingdom data (96.6%), possibly due to the low frequency of symptoms in India (*Appendix 1—table 9*). The corresponding frequency during the *Omicron period* was 81.5% (N=1,702).

### Epidemiology of Omicron variant in Laos

Population-level variant data from Laos were not available in the Global Initiative on Sharing All Influenza Data (GISAID) platform that covered the period between October 2021 and February 2022, and for this reason clinical data from this country were not included in analyses presented in the *Results* section of the manuscript. Local data suggest that Omicron variant spread in the country only after this period. Indeed, unpublished data from the Lao-Oxford-Mahosot Hospital-Wellcome Trust Research Unit indicate that Omicron variant was responsible for a large proportion of infections in March but not February 2022, although the numbers of infections genotyped were limited (Elizabeth Ashley, personal communication).

### Clinical data from Pakistan

In Pakistan, there was an increase in the relative frequency of Omicron variant during the period from October 2021 to February 2022. However, despite causing 96.1% of infections in the GISAID data from the country in mid-January 2022, throughout February this percentage fluctuated. Data from Pakistan were thus not included in the *Results* section. Here, we discuss clinical data from this country; for that, we used as the start of the *Omicron period* the first week when this variant was responsible for more than 90% of infections, regardless of whether this percentage was lower in the following weeks.

Data from 929 patients from Pakistan were contributed to the study; 249 records were from the *pre-Omicron period*, and 478, from the *Omicron period*. The percentage of patients with at least one symptom was 83.9% in the *pre-Omicron period*, and 57.9%, in the *Omicron period*. 52.2% and 59.2% had at least one comorbidity during these two periods, respectively. Vaccination data were available for 474 patients admitted during the study periods: 37.7% and 62.9% had history of COVID-19 vaccination during the *pre-Omicron* and *Omicron periods*. The 14-day fatality risk for hospitalised patients during the *pre-Omicron period* was 52.5%, and during the *Omicron period*, 45.4%.

### Sensitivity analysis that excludes patients with other primary reason for hospitalisation

For 30,052 patients admitted during the two study periods, information was available on whether COVID-19 was the primary medical reason for hospitalisation; most of these patients were from South Africa. As a sensitivity analysis, we fit a mixed-effects logistic regression model on the 14-day fatality risk excluding patients who had reported that COVID-19 was not the reason for hospitalisation; patients for whom this information was missing were included. The odds ratio for the association between study period and 14-day fatality risk was 0.68 (95% confidence interval 0.61–0.75).

**Appendix 1—table 1.** Numbers of records contributed by partner institutions in different countries between 01/10/2021 and 28/02/2022.

| Country | Number of records |
|---|---|
| South Africa | 69766 |
| United Kingdom | 55049 |
| Pakistan | 929 |
| Canada | 919 |
| Nepal | 504 |
| Laos | 456 |
| India | 409 |
| Romania | 166 |
| Saudi Arabia | 151 |
| Spain | 151 |
| Netherlands | 134 |
| Malaysia | 90 |
| Norway | 67 |
| Turkey | 57 |
| Brazil | 54 |
| Colombia | 52 |
| New Zealand | 46 |
| Kuwait | 35 |
| United States | 32 |
| Philippines | 26 |
| Ghana | 21 |
| Ireland | 20 |
| Israel | 17 |
| Italy | 12 |
| Estonia | 7 |
| Australia | 7 |
| Indonesia | 6 |
| Portugal | 5 |
| Germany | 4 |
| Argentina | 4 |

**Appendix 1—table 2.** Missing data on symptoms.
Note that this information was not systematically recorded in South Africa, and for this reason data from that country are not included in this table.

| Symptoms | Yes | No | Missing data |
|---|---|---|---|
| Any cough | 20431 | 13726 | 25273 |
| Fever | 16045 | 19465 | 23920 |
| Headache | 3896 | 28398 | 27136 |

*Appendix 1—table 2 Continued on next page*

*Appendix 1—table 2 Continued*

| Symptoms | Yes | No | Missing data |
|---|---|---|---|
| *Confusion* | 5960 | 28548 | 24922 |
| *Seizures* | 570 | 33424 | 25436 |
| *Sore throat* | 2394 | 29353 | 27683 |
| *Runny nose* | 1639 | 30279 | 27512 |
| *Vomiting* | 6956 | 27734 | 24740 |
| *Wheezing* | 2042 | 31191 | 26197 |
| *Diarrhoea* | 4418 | 29989 | 25023 |
| *Chest pain* | 5488 | 28732 | 25210 |
| *Conjunctivitis* | 106 | 32606 | 26718 |
| *Myalgia* | 3686 | 28195 | 27549 |
| *Rash* | 476 | 32877 | 26077 |
| *Fatigue* | 11339 | 22150 | 25941 |
| *Ageusia* | 1682 | 28341 | 29407 |
| *Inability to walk* | 252 | 3797 | 55381 |
| *Anosmia* | 1393 | 29040 | 28997 |
| *Shortness of breath* | 20490 | 14030 | 24910 |
| *Lymphadenopathy* | 145 | 32795 | 26490 |

**Appendix 1—table 3.** Missing data on comorbidities. In this table, data from all countries are included.

| Comorbidities | Yes | No | Missing data |
|---|---|---|---|
| *Liver disease* | 1786 | 40992 | 86418 |
| *Diabetes* | 12956 | 68743 | 47497 |
| *Chronic cardiac disease* | 13546 | 73423 | 42227 |
| *Hypertension* | 32052 | 57401 | 39743 |
| *Current smoking* | 5090 | 26674 | 97432 |
| *COPD* | 9304 | 77794 | 42098 |
| *Active TB* | 1579 | 45731 | 81886 |
| *Asthma* | 8720 | 79175 | 41301 |
| *Chronic kidney disease* | 8441 | 78453 | 42302 |
| *Malignant neoplasm* | 5062 | 81465 | 42669 |
| *Dementia* | 4646 | 38530 | 86020 |
| *HIV* | 5925 | 79121 | 44150 |
| *Chronic neurological disorder* | 5615 | 37740 | 85841 |
| *Obesity* | 5723 | 45367 | 78106 |

**Appendix 1—table 4.** Potential limitations of population-level variant data used to determine time periods when Omicron variant was dominant.

| Potential limitation | Likely impact on analyses |
|---|---|
| Population-level data come from a range of sources in each country, and for most samples it is not possible to determine whether patient was hospitalised or was a community (mild) case | If different variants are associated with different severities upon infection and if a large fraction of samples used in the estimation of population-level frequency of variants are from community cases, then it is possible that this frequency does not fully represent the frequency in the hospitalised population. In particular, if Omicron variant infection is linked to lower risk of hospitalisation, as previous studies suggest, it is possible that even during periods when community-level frequency of Omicron variant was high, the frequency of Omicron variant in the hospitalised population might have been relatively low. |
| Use of country-level data, rather than data on variant frequency in the catchment areas of clinical centres contributing data | If Omicron variant spreads asynchronously in a country, with some regions reaching high relative frequency faster than others, it is possible that country-level data, rather than data at a finer geographical level, might not reflect Omicron variant frequency in the population from which patients were recruited. |
| Delay between infection, onset of symptoms and hospitalisation | Depending on the data source used to define population-level frequency of variants, if clinical samples were obtained early during the infection, hospitalised cases might only have the same variant composition after a time lag, corresponding to average time from infection, or onset of symptoms, to hospital admission. |

**Appendix 1—table 5.** Numbers of records in the *pre-Omicron* and *Omicron periods* by country.

| | Omicron emergence | | |
|---|---|---|---|
| Country | Before 10% | After 90% | Total |
| *South Africa* | 4180 | 51929 | 56109 |
| *United Kingdom* | 18124 | 26479 | 44603 |
| *Canada* | 61 | 763 | 824 |
| *Nepal* | 197 | 204 | 401 |
| *India* | 89 | 212 | 301 |
| *Netherlands* | 60 | 65 | 125 |
| *Saudi Arabia* | 2 | 121 | 123 |
| *Romania* | 1 | 100 | 101 |
| *Spain* | 21 | 56 | 77 |
| *Malaysia* | 42 | 11 | 53 |
| *Norway* | 5 | 45 | 50 |
| *Brazil* | 15 | 33 | 48 |
| *New Zealand* | 34 | 6 | 40 |
| *Colombia* | 26 | 5 | 31 |
| *Turkey* | 0 | 27 | 27 |
| *Philippines* | 16 | 5 | 21 |
| *United States of America* | 14 | 7 | 21 |
| *Kuwait* | 0 | 19 | 19 |
| *Ghana* | 4 | 15 | 19 |
| *Ireland* | 14 | 3 | 17 |
| *Israel* | 0 | 14 | 14 |

*Appendix 1—table 5 Continued on next page*

*Appendix 1—table 5 Continued*

|  | **Omicron emergence** | | |
|---|---|---|---|
| Australia | 0 | 6 | 6 |
| Portugal | 3 | 2 | 5 |
| Indonesia | 1 | 4 | 5 |
| Germany | 2 | 2 | 4 |
| Italy | 3 | 0 | 3 |
| Argentina | 0 | 3 | 3 |
| Estonia | 1 | 0 | 1 |

**Appendix 1—table 6.** Medians (interquartile ranges [Q1 - Q3]) of age by study period and country. Only countries with 10 or more observations in both study periods are shown.

| | **Before 10%** | | | **After 90%** | | |
|---|---|---|---|---|---|---|
| Country | Median | Q1 | Q3 | Median | Q1 | Q3 |
| Brazil | 59 | 50 | 70 | 55 | 48 | 70 |
| Canada | 63 | 50 | 71 | 62 | 35 | 76 |
| Spain | 68 | 63 | 75 | 76 | 59 | 84 |
| United Kingdom | 66 | 48 | 78 | 67 | 38 | 81 |
| India | 63 | 47 | 72 | 70 | 60 | 76 |
| Malaysia | 63 | 52 | 68 | 59 | 55 | 63 |
| Netherlands | 74 | 64 | 80 | 70 | 55 | 77 |
| Nepal | 63 | 42 | 77 | 64 | 42 | 75 |
| South Africa | 45 | 30 | 62 | 41 | 27 | 63 |

**Appendix 1—table 7.** Numbers of hospitalised patients admitted due to COVID-19. For country-time period combinations with less than 10 observations, numbers are not presented.

| | **Before 10%** | | | **After 90%** | | |
|---|---|---|---|---|---|---|
| Country | COVID-19 as reason (N) | COVID-19 as reason (%) | Total | COVID-19 as reason (N) | COVID-19 as reason (%) | Total |
| Australia |  |  |  | - | - | <10 |
| Argentina |  |  |  | - | - | <10 |
| Brazil | 14 | 100 | 14 | 32 | 96.7 | 33 |
| Canada | 12 | 52.2 | 23 | 514 | 67.5 | 761 |
| Colombia | 2 | 7.7 | 26 | - | - | <10 |
| Germany | - | - | <10 | - | - | <10 |
| Ghana | - | - | <10 |  |  |  |
| India | 0 | 0 | 12 | 2 | 8.3 | 24 |
| Indonesia |  |  |  | - | - | <10 |
| Israel |  |  |  | 8 | 66.7 | 12 |
| Kuwait |  |  |  | 0 | 0 | 18 |
| Malaysia | - | - | <10 | - | - | <10 |
| Nepal | - | - | <10 | 0 | 0 | 15 |
| Netherlands | 49 | 81.7 | 60 | 39 | 61.9 | 63 |

*Appendix 1—table 7 Continued on next page*

*Appendix 1—table 7 Continued*

|  | Before 10% |  |  | After 90% |  |  |
|---|---|---|---|---|---|---|
| New Zealand | 30 | 90.9 | 33 | - | - | <10 |
| Norway | - | - | <10 | 34 | 75.5 | 45 |
| Philippines | 16 | 100 | 16 | - | - | <10 |
| Romania | - | - | <10 | 100 | 100 | 100 |
| Saudi Arabia | - | - | <10 | 68 | 68.7 | 99 |
| South Africa | 1433 | 71.1 | 2015 | 18306 | 69.0 | 26512 |
| Spain | 11 | 64.7 | 17 | 37 | 66.1 | 56 |
| Turkey |  |  |  | 27 | 100 | 27 |
| USA | 0 | 0 | 11 | - | - | <10 |

**Appendix 1—table 8.** Percentages of patients with at least one comorbidity by country and study period.
Only countries with at least 10 patients in each study period are included.

|  | Before 10% |  | After 90% |  |
|---|---|---|---|---|
| Country | % with one or more comorbidities | Total | % with one or more comorbidities | Total |
| Brazil | 78.6 | 14 | 81.8 | 33 |
| Canada | 76.7 | 60 | 74.2 | 760 |
| India | 44.9 | 89 | 56.9 | 209 |
| Malaysia | 64.3 | 42 | 72.7 | 11 |
| Nepal | 46.2 | 197 | 55.4 | 204 |
| Netherlands | 86.7 | 60 | 78.5 | 65 |
| South Africa | 53.9 | 3170 | 44.7 | 37412 |
| Spain | 76.2 | 21 | 76.8 | 56 |
| United Kingdom | 86.5 | 11820 | 84.6 | 21501 |

**Appendix 1—table 9.** Percentages of patients with at least one symptom by country and study period.
Only countries with at least 10 patients in each study period are included.

|  | Before 10% |  | After 90% |  |
|---|---|---|---|---|
| Country | % with one or more symptoms | Total | % with one or more symptoms | Total |
| Brazil | 100.0 | 14 | 100.0 | 32 |
| Canada | 91.7 | 60 | 91.6 | 754 |
| India | 28.1 | 89 | 18.6 | 210 |
| Malaysia | 64.3 | 42 | 90.9 | 11 |
| Nepal | 97.0 | 197 | 86.3 | 204 |
| Netherlands | 96.7 | 60 | 96.9 | 65 |
| Spain | 100.0 | 21 | 87.5 | 56 |
| United Kingdom | 97.4 | 11104 | 89.3 | 16157 |

**Appendix 1—table 10.** Medians (interquartile ranges [Q1 - Q3]) of time from admission or disease onset to death by study period and country.

Only countries with 10 or more observations in both study periods are presented.

| | *pre-Omicron period* | | | *Omicron period* | | |
|---|---|---|---|---|---|---|
| Country | Median | Q1 | Q3 | Median | Q1 | Q3 |
| *Canada* | 10 | 6 | 21 | 10 | 5 | 18 |
| *United Kingdom* | 11 | 6 | 19 | 11 | 6 | 19 |
| *India* | 6 | 3 | 8 | 7 | 3 | 12 |
| *Nepal* | 6 | 5 | 12 | 4 | 2 | 8 |
| *South Africa* | 6 | 2 | 12 | 5 | 2 | 10 |

**Appendix 1—table 11.** Survival models.

Results of a Cox proportional hazards model, stratified by country, on time to death in the first 28 days since hospital admission or onset of symptoms, which happened latest, are shown in the Hazard ratio column. For this analysis, if follow-up duration was longer than 28 days, it was set to 28 days, and patients who were discharged were censored on the day of discharge. The assumption of proportional hazards was violated for the variable on previous vaccination; for this reason, the model was also stratified by this variable. An alternative analysis assumed that patients discharged from hospital were censored on day 28; in this analysis, the hazard ratio for the variable corresponding to study period was 0.68 (0.63–0.74); for this model, the proportional hazards assumption did not hold for the study period variable. We also fit a competing risk model, with hospital discharge as competing event; estimates from this model are presented in the Subhazard ratio column. In this model, previous COVID-19 vaccination was included as a covariate (subhazard ratio 0.55, 95% CI 0.52–0.59). We also fit a competing risk model using only data from the six countries included in *Figures 3–5* and that included country as a dummy variable; in this model, the subhazard ratio for the Omicron period variable was 0.68 95% CI (0.63–0.74).

| | | Hazard ratio | Subhazard ratio |
|---|---|---|---|
| **Variables** | | | |
| | *Omicron period* | 0.77 (0.71–0.84) | 0.79 (0.73–0.84) |
| | *Sex (male)* | 1.24 (1.17–1.32) | 1.32 (1.24–1.40) |
| | *Age* | | |
| | *Older than 60 years* | Reference | |
| | *Aged between 18 and 60 years* | 0.41 (0.38–0.44) | 0.26 (0.24–0.28) |
| | *Younger than 18 years* | 0.13 (0.11–0.17) | 0.06 (0.04–0.07) |

**Appendix 1—table 12.** Distribution of infections with individual-level variant information by country and variant.

Only countries with at least 10 observations for Delta and Omicron variants are listed. Note that other countries had limited numbers for both or one of the two variants.

| Country | Delta | Omicron |
|---|---|---|
| *Canada* | 26 | 303 |
| *Netherlands* | 12 | 52 |
| *Norway* | 15 | 22 |
| *South Africa* | 17 | 720 |
| *Spain* | 10 | 16 |

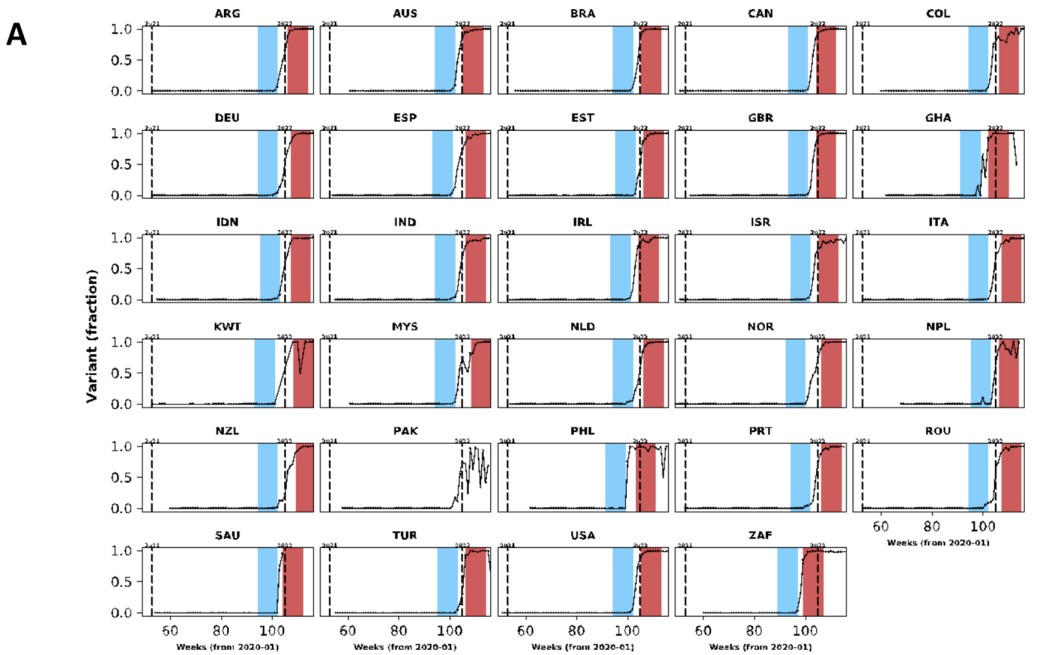

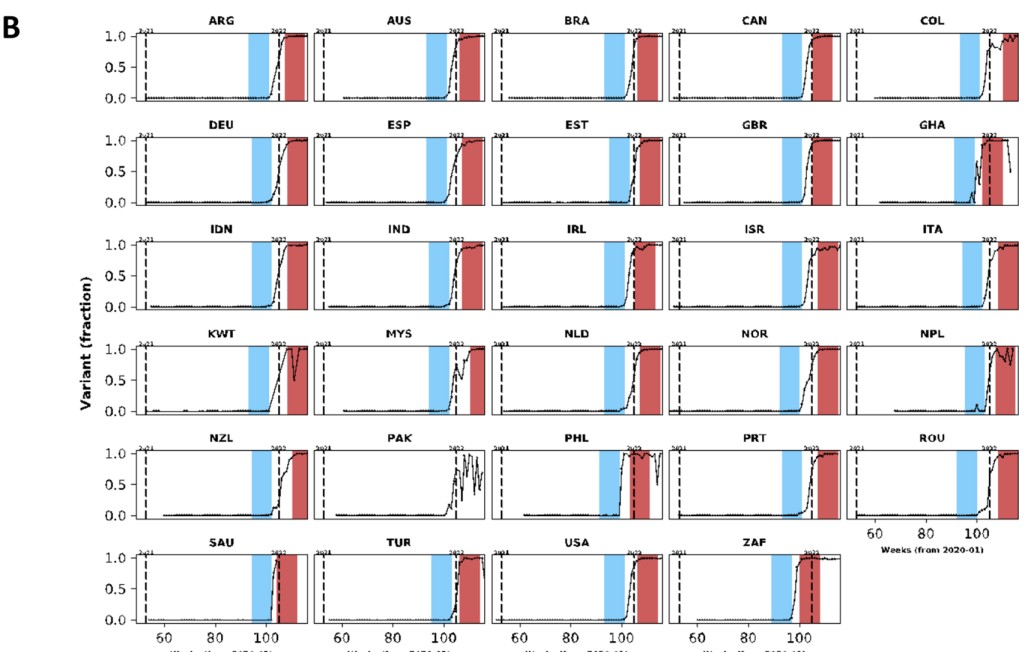

**Appendix 1—figure 1.** In this figure, population-level variant data are presented for countries with clinical data included in our analysis. The same structure of *Figure 1* was used but different cut-off frequencies for Omicron variant were applied: in (**A**), the lower and upper threshold frequencies were 10% and 80%; in (**B**), these frequencies were 5% and 90%.

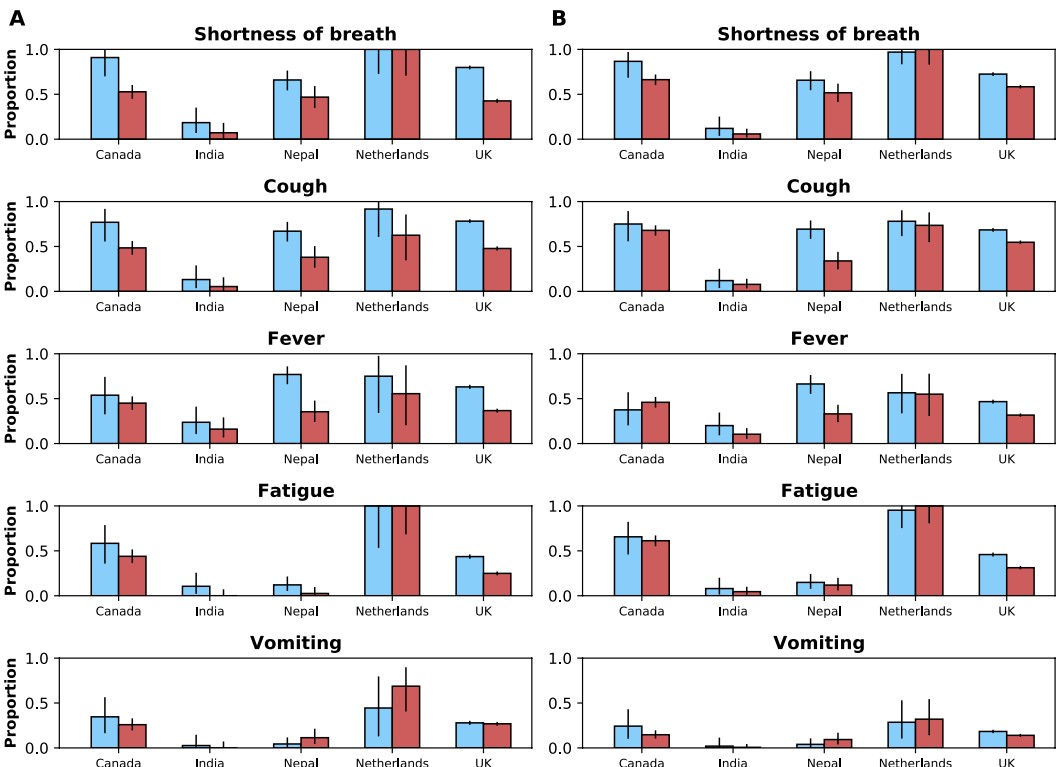

**Appendix 1—figure 2.** Frequencies of the five most common symptoms during the period before (blue bars) and after (red bars) Omicron variant frequency reached 10% and 90%, respectively. 95% confidence intervals are also shown. In (**A**), data from individuals aged between 18 and 60 years are shown; and (**B**) shows the same information for individuals older than 60 years. Data from children are not presented.

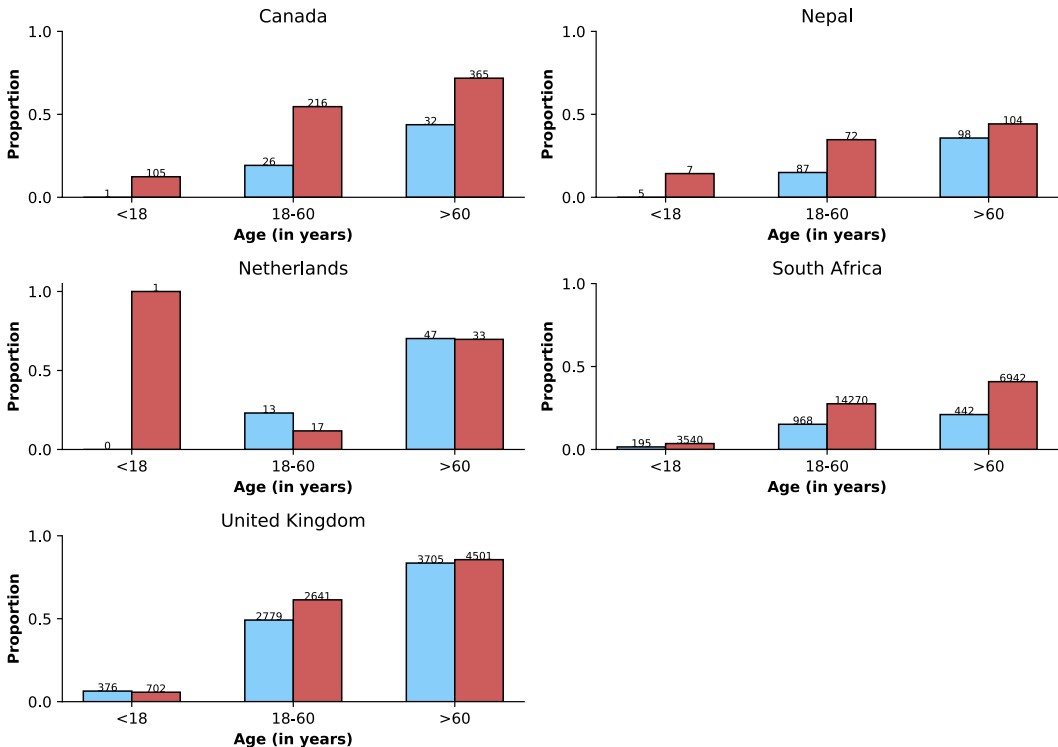

**Appendix 1—figure 3.** Frequency of previous vaccination by study period, age category and country. Only data from countries with at least 50 observations with information on previous vaccination during both study periods defined by Omicron variant frequency are shown. In each panel, the x-axis shows different age categories, with blue bars corresponding to the *pre-Omicron period* and red bars, to the period after Omicron variant frequency, relative to other variants, reaches 90%. Above each bar, the total number of records included in the calculation of the proportions (y-axes) are presented.

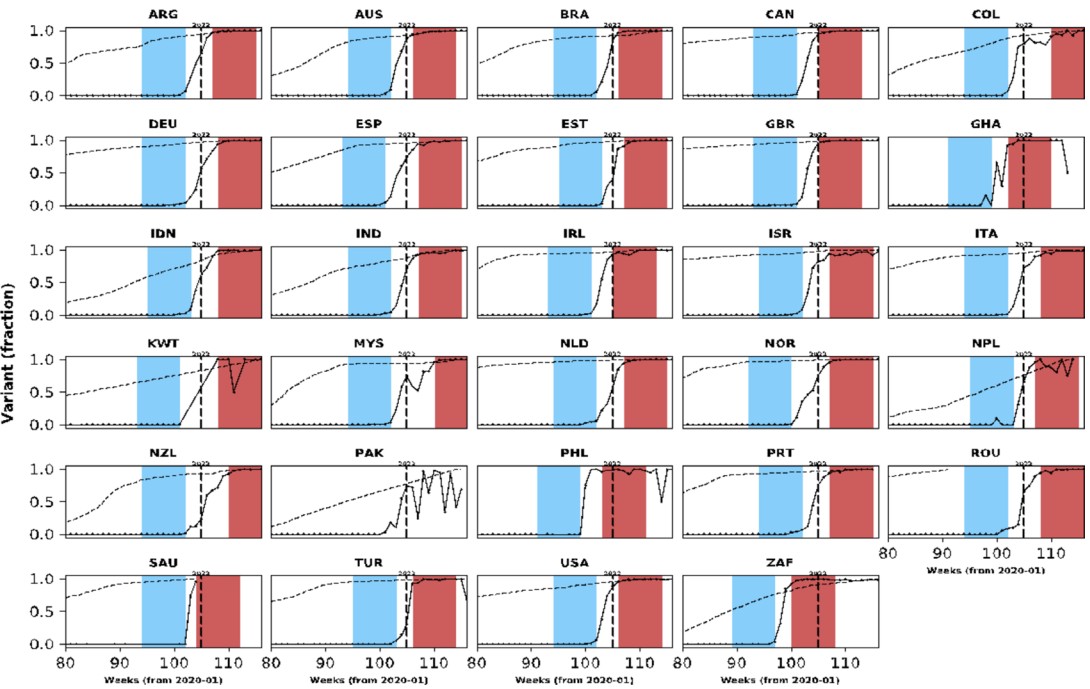

**Appendix 1—figure 4.** Population-level vaccination coverage. Data from different countries are presented in different panels; x-axes show epidemiological weeks since the first epidemiological week of 2020. As in *Figure 1*, *Appendix 1—figure 4 continued on next page*

*Appendix 1—figure 4 continued*
continuous black lines represent frequency of Omicron variant relative to the other variants. In addition to information on Omicron variant frequency, each panel also shows data on vaccination: the dashed line shows the proportion of population vaccinated with at least one dose relative to the maximum number vaccinated in each country at the time of the analysis (March 2022). Data used to generate this figure were downloaded from https://ourworldindata.org/.

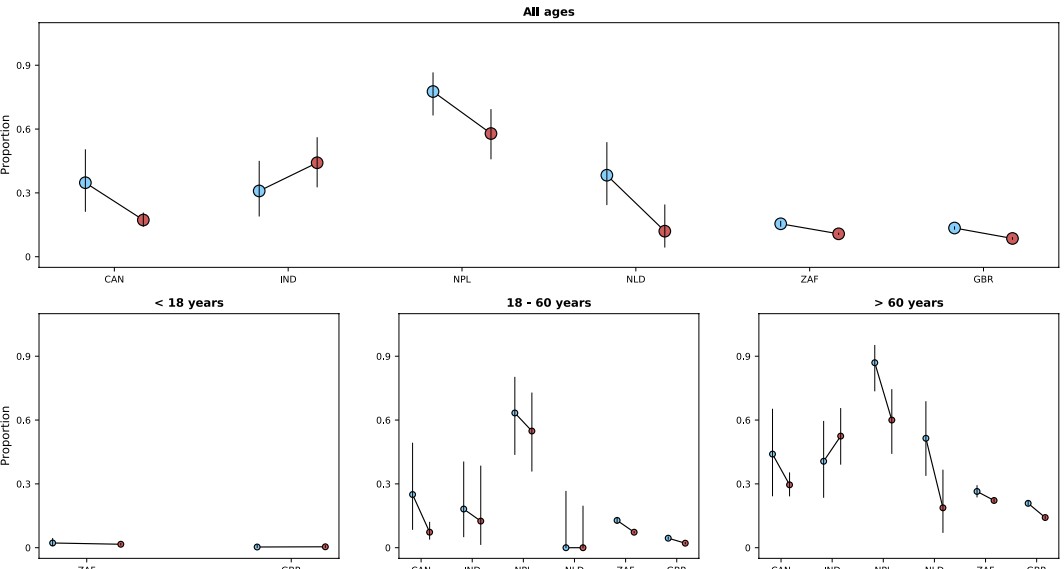

**Appendix 1—figure 5.** Risk of death in the first 28 days after hospital admission or disease onset, whichever occurred latest, during *pre-Omicron* and *Omicron periods*. In each panel, the x-axis shows countries, with different periods represented by circles with different colours (blue circles for the *pre-Omicron period*; red circles, for period after Omicron variant frequency reaches 90%). 95% confidence intervals are presented. The top panel shows data for individuals of all ages; the bottom panels, data for patients aged less than 18 years, between 18 and 60 years, and older than 60 years. Only countries with at least 50 observations in both study periods are included in the figure; for panels presenting age-specific estimates (bottom row), a further requirement for inclusion was outcome data for at least 10 patients in the corresponding age range in both periods.

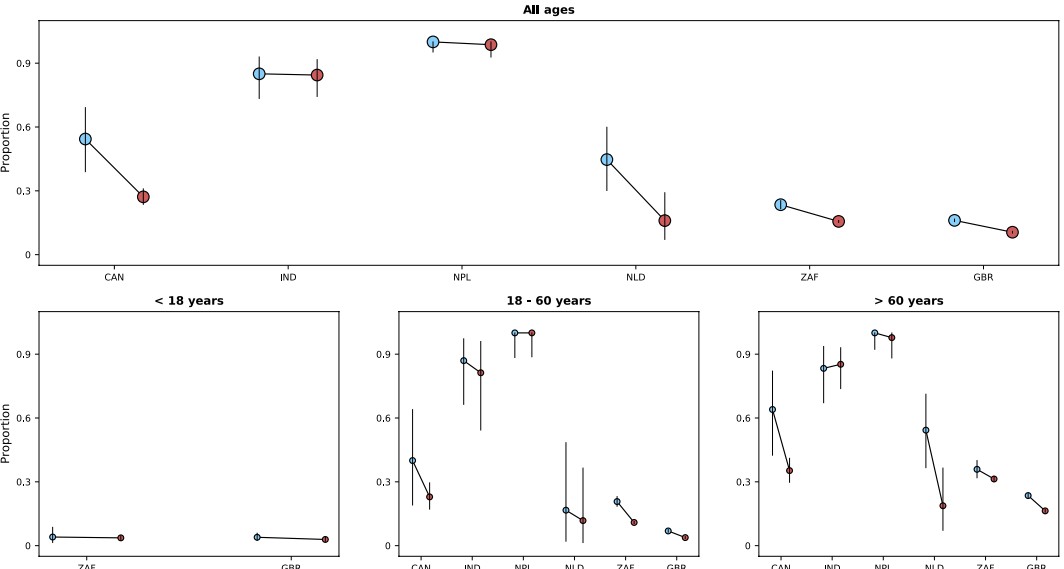

**Appendix 1—figure 6.** Risk of death or invasive mechanical ventilation by study period. In each panel, the x-axis shows countries, with different periods represented by circles with different colours (blue circles for the *pre-Omicron period*; red circles, for the *Omicron period*). 95% confidence intervals are presented. The top panel shows data for individuals of all ages; the bottom panels, data for patients aged less than 18 years, between 18 and 60 years, and older than 60 years. Only countries with at least 50 observations in both study periods are included in the figure; for panels presenting age-specific estimates (bottom row), a further requirement for inclusion was outcome data for at least 10 patients in the corresponding age range in both periods. Different from *Figure 5* and *Appendix 1—figure 5*, time since hospital admission or onset of symptoms was not used since for most patients who required invasive mechanical ventilation the start date of the therapeutic approach was not available. Only patients with information on invasive mechanical ventilation use and who were either discharged or died were included.

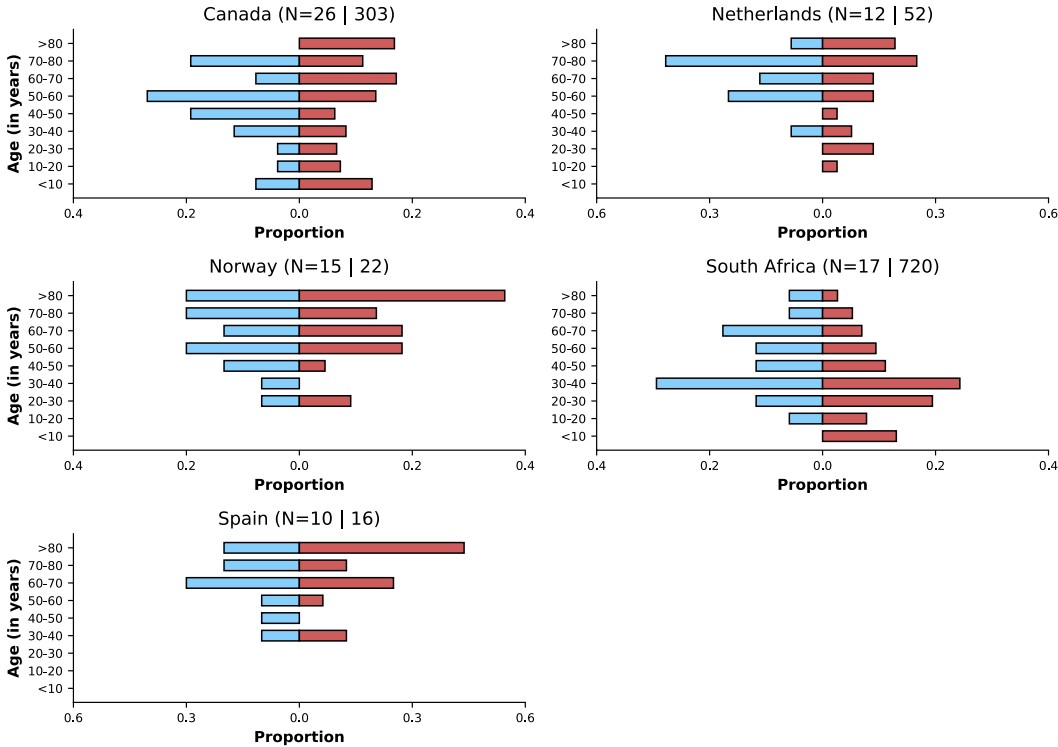

**Appendix 1—figure 7.** This figure shows similar information to that presented in *Figure 3*. The legend of that figure applies to this figure, except that instead of referring to time periods, the figure shows data for Delta and Omicron variants. Only countries with at least 10 observations for Delta and Omicron variants are included.

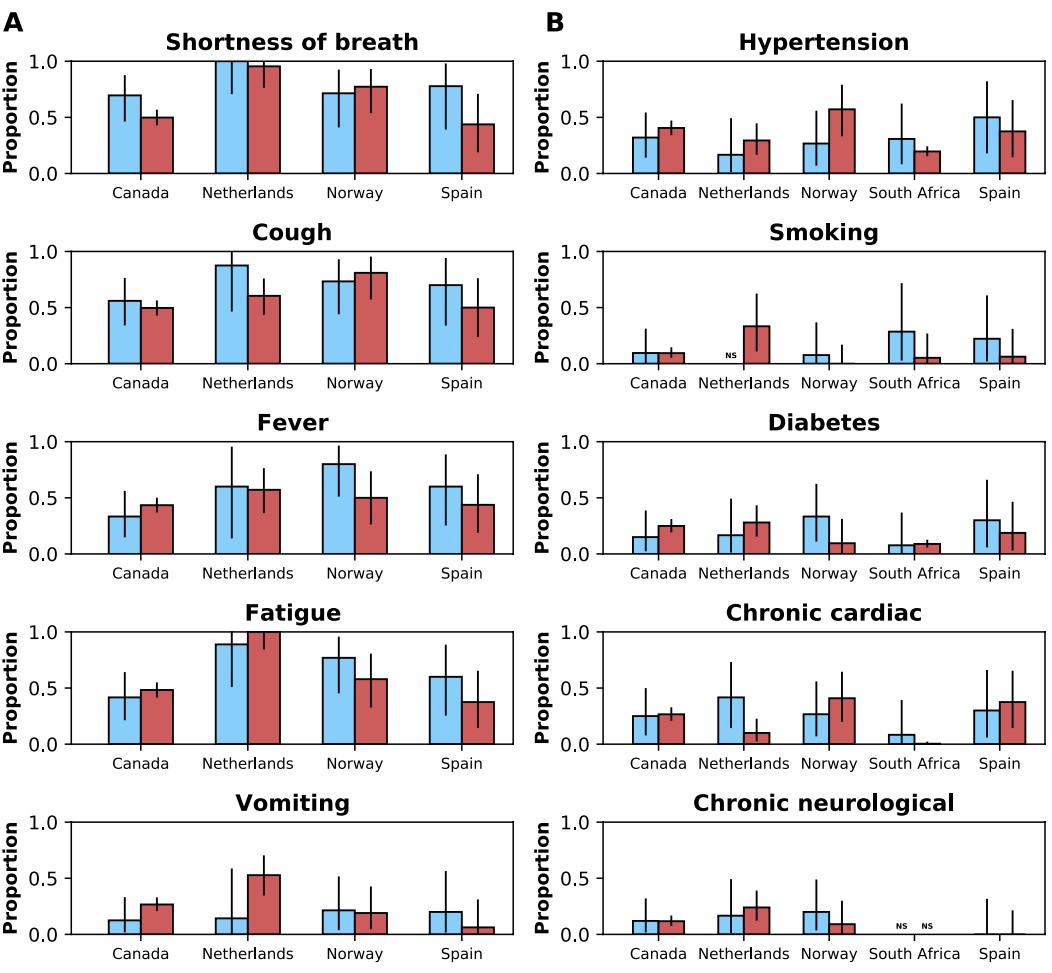

**Appendix 1—figure 8.** This figure shows similar information to that presented in *Figure 4*. The legend of that figure applies to this figure, except that instead of referring to time periods, the figure shows data for Delta and Omicron variants. Only countries with at least 10 observations for Delta and Omicron variants are included.

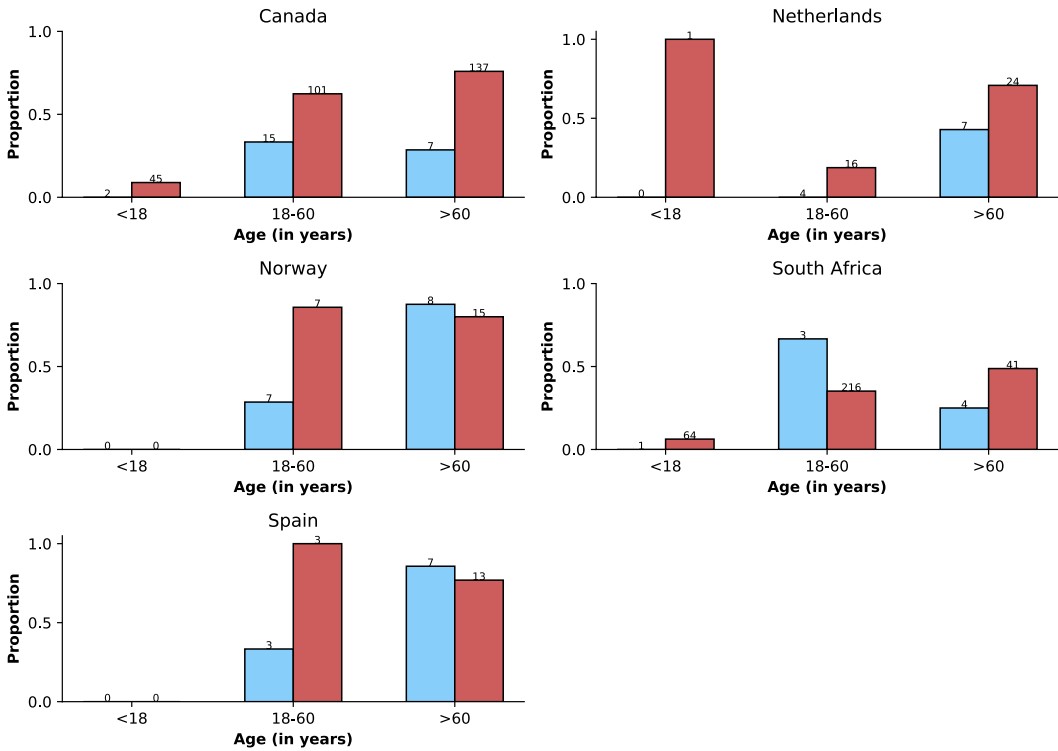

**Appendix 1—figure 9.** This figure shows similar information to that presented in *Appendix 1—figure 3*. The legend of that figure applies to this figure, except that instead of referring to time periods, the figure shows data for Delta and Omicron variants. Only countries with at least 10 observations for Delta and Omicron variants are included; note that, different from *Appendix 1—figure 3*, the criterion did not consider missingness of vaccination data.

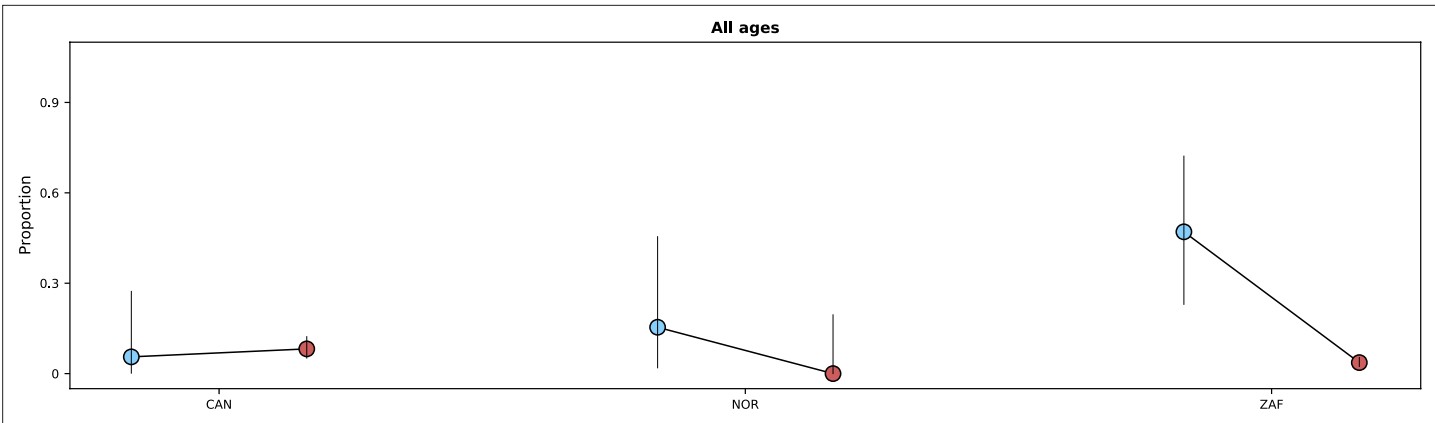

**Appendix 1—figure 10.** This figure shows similar information to that presented in *Figure 5*. The legend of that figure applies to this figure, except that instead of referring to time periods, the figure shows data for Delta and Omicron variants. Only countries with at least 10 observations for Delta and Omicron variants are included. Age-stratified panels are not shown due to the limited number of observations with individual-level variant data.

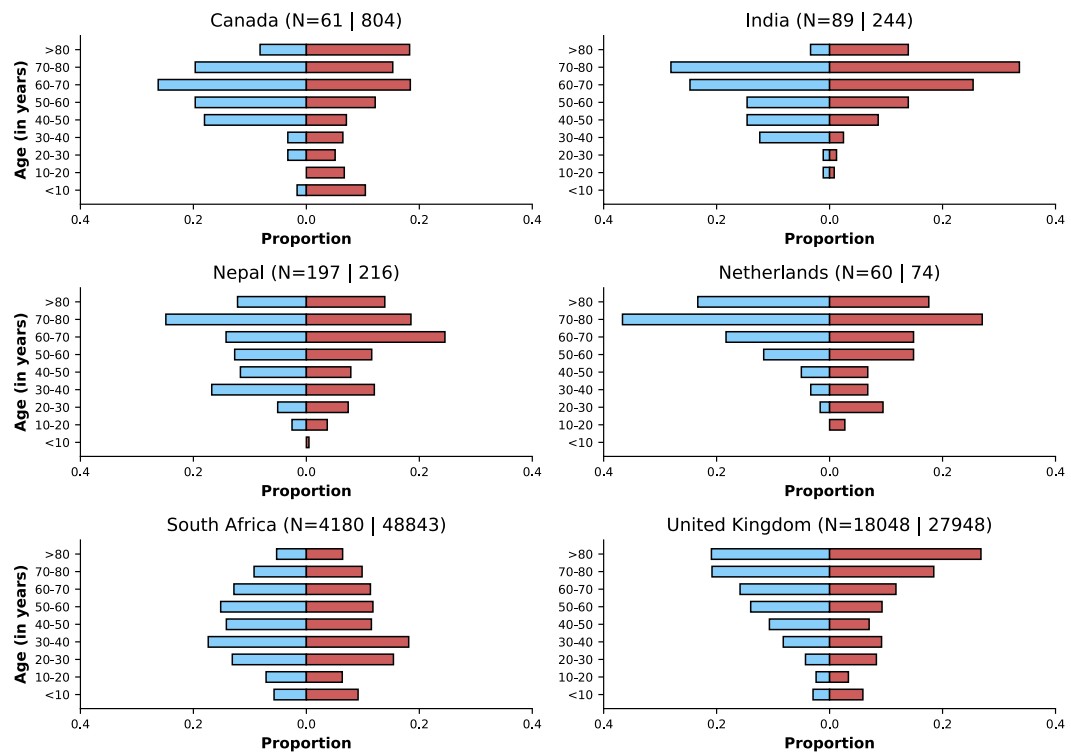

**Appendix 1—figure 11.** This figure shows similar information to that presented in *Figure 3*. The legend of that figure applies to this figure. Here, the upper threshold frequency used to define Omicron variant dominance was 80% rather than 90%.

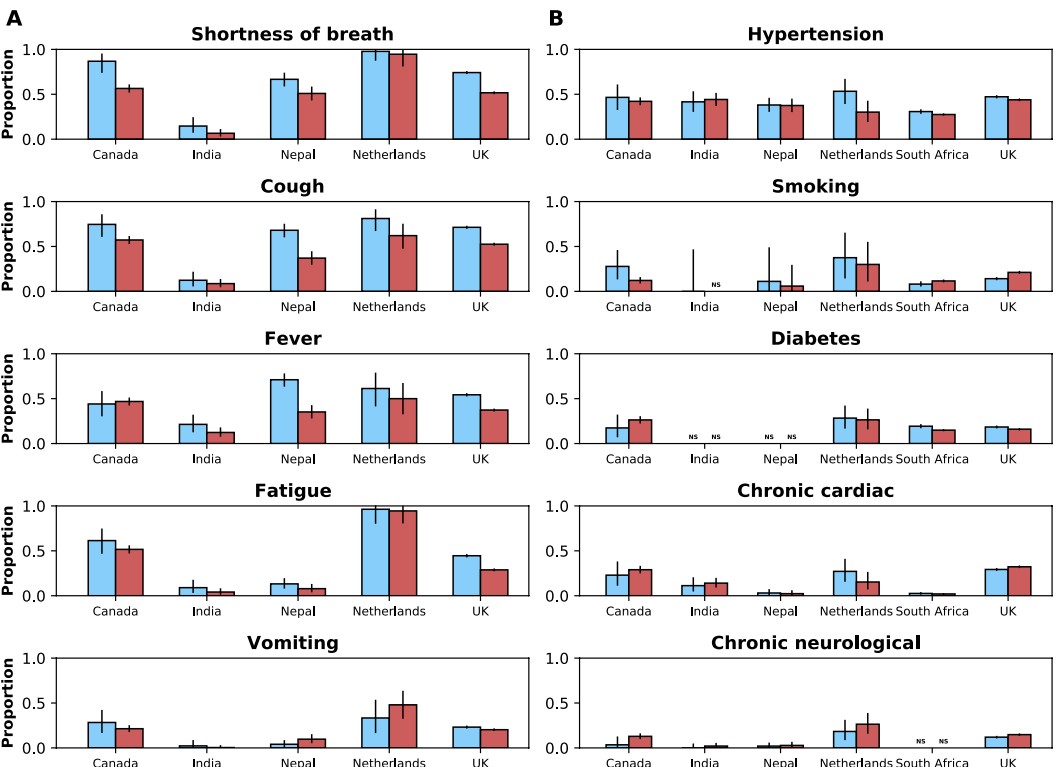

**Appendix 1—figure 12.** This figure shows similar information to that presented in *Figure 4*. The legend of that figure applies to this figure. Here, the upper threshold frequency used to define Omicron variant dominance was 80% rather than 90%.

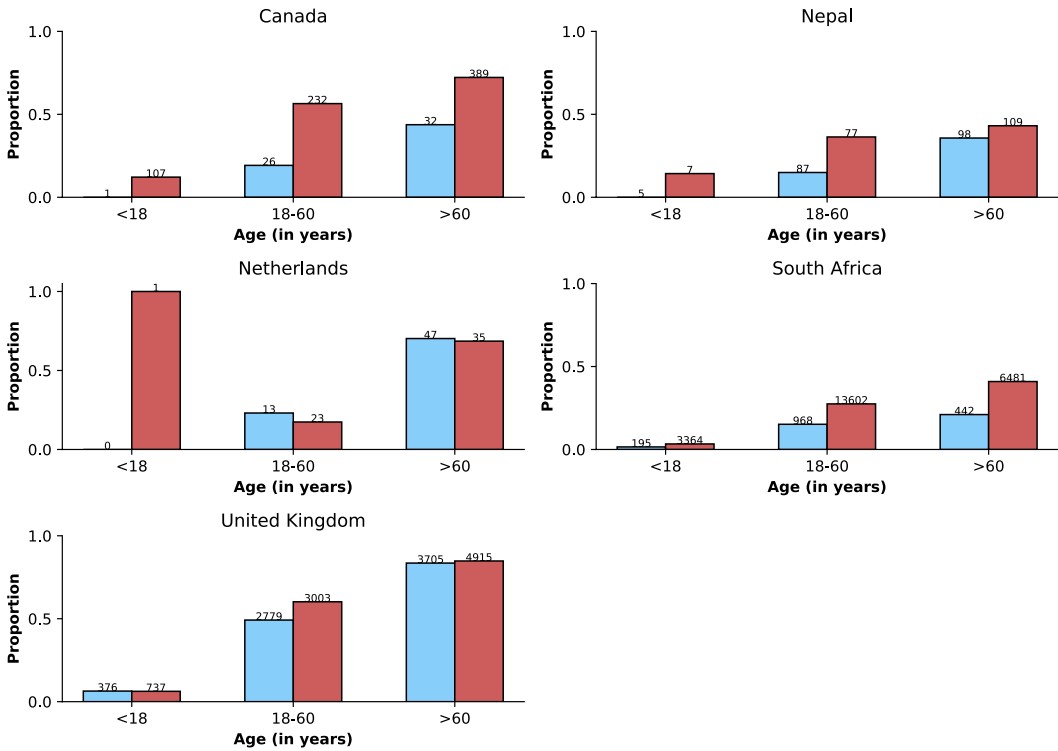

**Appendix 1—figure 13.** This figure shows similar information to that presented in *Appendix 1—figure 3*. The legend of that figure applies to this figure. Here, the upper threshold frequency used to define Omicron variant dominance was 80% rather than 90%.

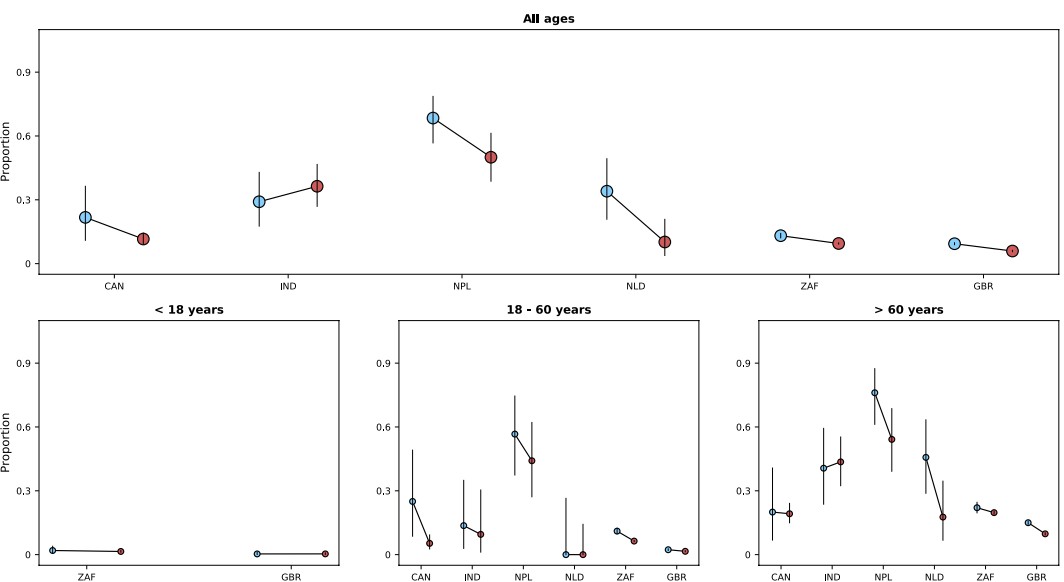

**Appendix 1—figure 14.** This figure shows similar information to that presented in *Figure 5*. The legend of that figure applies to this figure. Here, the upper threshold frequency used to define Omicron variant dominance was 80% rather than 90%.

