## [Editor Report]

This manuscript compares COVID-19 mortality during the pre-Omicron and Omicron emergence periods in several countries. It finds evidence suggesting the Omicron variant was associated with lower mortality than previous dominant variants in multiple countries, though other factors than changing variant virulence might explain these observations, as discussed by the authors. This paper will be of interest to infectious disease scientists both for its content and its methods, as it validates that population-level variant frequency can be a good proxy for individual-level variant data to derive insights on variant biology with population data.

---

## [Decision Letter]

**Decision letter after peer review:**

Thank you for submitting your article "An international observational study to assess the impact of the Omicron variant emergence on the clinical epidemiology of COVID-19 in hospitalised patients" for consideration by *eLife*. Your article has been reviewed by 3 peer reviewers, one of whom is a member of our Board of Reviewing Editors, and the evaluation has been overseen by a Senior Editor. The following individual involved in review of your submission has agreed to reveal their identity: Matthew Whitaker (Reviewer #3).

As is customary in *eLife*, the reviewers have discussed their critiques with one another. What follows below is the Reviewing Editor's edited compilation of the essential and ancillary points provided by reviewers in their critiques and in their interaction post-review. Please submit a revised version that addresses these concerns directly. Although we expect that you will address these comments in your response letter, we also need to see the corresponding revision clearly marked in the text of the manuscript. Some of the reviewers' comments may seem to be simple queries or challenges that do not prompt revisions to the text. Please keep in mind, however, that readers may have the same perspective as the reviewers. Therefore, it is essential that you attempt to amend or expand the text to clarify the narrative accordingly.

Essential revisions:

1. Please address the various requests for clarifications brought up by reviewers #1 and #2, which will help readers better understand the methods of the paper, especially in regard to inclusion/exclusion criteria and recruitment of patients.

2. Please consider performing a sensitivity analysis restricted to countries with large numbers of cases as suggested by reviewer #2 to assess whether results were impacted by the inclusion of countries with few cases; it is likely that the cases from these countries are highly selected and therefore that their inclusion may have led to a selection bias.

*Reviewer #1 (Recommendations for the authors):*

• Figure 2: it is not clear to me why there were 12,665 records excluded from before or after the study periods, as to my mind these would have fit the definition of pre- and post- Omicron. Please explain.

• Table S7: please add percentages of patients hospitalized for COVID as well as numbers as these are easier to compare between countries and periods.

• Table S11: This analysis shows the results from a standard Cox model where discharge is treated as a censored observation. However, discharge is informative censoring, as presumably, the risk of death is much lower in a discharged patient. The authors may want to consider performing a competing risk analysis (Fine & Grey model) instead of a Cox model for this sensitivity analysis. The results would be more comparable to the logistic regressions, where discharges are included in the denominators but not the numerators.

• Figure S4: it is not clear how vaccination coverage is defined here (1 dose? 2 doses? 3 doses?)

• The authors may want to comment on the results from individual countries where mortality increased in the Omicron era (ex. India, Brazil, Spain), and the factors that might explain this. Presumably, these results may be due to some confounding by age and clinical profile of patients in both eras.

*Reviewer #2 (Recommendations for the authors):*

This is a great attempt for collaborative work, and we do know and appreciate how difficult it is to set up such a framework.

However, when reading the paper, it looks as if the project is not mature yet. Patients' enrolment per country is extremely disproportional and discredits the project. Can you just include the countries that sent at least 100 patients's data and name all countries as part of the consortium? countries that have sent very few forms should not be mentioned.

Patients' data sources seem to differ a lot per country, at the end it looks like a patchwork of random data/patients that is being analyzed to see if it could be of any use.

To make the paper clearer I would suggest that you:

1. Clarify the goals:

If the goal is to assist policy makers when a new variant emerges it will always be too late as variants are not introduced in different countries on the same calendar week. For Omicron (a bad example) policy makers reacted within 24h or less after the first announcement of this new variant.

Such collaborative programme could be very useful to identify unique variant characteristics compared to previous variants and identify specific country patterns linked to different variant exposure, different vaccination level, comorbidity rates, access to care etc…such global initiative doesn't exist yet.

2. Clarify which patients are included in the data base. All COVID-19 patients hospitalized, only the ones with COVID-19 as main reason for hospitalization, critical care patients, all patients with respiratory symptoms? How are patients with COVID-19 as the main reason for hospitalization?

3. Standardize the data so that the analysis is not done on different populations for each criteria. Currently symptoms analysis seems to mostly reflect UK patients, vaccination status mostly South Africa patients.

4. Again I would keep the top 8 or 10 countries, the small number of data provided by the other countries introduces confusion and alter the impact of your paper.

Other questions brought up by the reviewer to be addressed by authors (editor's note: originally these questions were in the public review but have been moved to the comments to authors to be addressed):

• The study is presented as a multi-center international study that includes more than 100,000 patients from 30 countries, however, 96.6% of the study patients originated from 2 countries, South Africa (54%) and the United Kingdom (42.6%). Can this still qualify as a multicenter study?

• Country specific medians suggest that the younger age of patients after the Omicron variant experience in the combined dataset is at least partially explained by an increase of data contributed by South Africa. Could the high proportion of South African patients' data have impacted on other measures too?

• Are the patients recruited COVID-19 proven patients? Are incidental cases included or excluded? How was the primary reason for hospitalization identified? Interpretation of mortality rate could be influenced by the recruitment of patients.

*Reviewer #3 (Recommendations for the authors):*

I only have one specific suggestion for further analysis:

In the Results section on vaccination history, it is implied that more granular detail on vaccination status of patients (ie number of vaccines received) is available but is not used in the models. If this is the case, it might be interesting to see a version of the main model adjusted on number of vaccine doses, even if this is restricted to countries where this data is available.

---

## [Author Response]

Essential revisions:1. Please address the various requests for clarifications brought up by reviewers #1 and #2, which will help readers better understand the methods of the paper, especially in regard to inclusion/exclusion criteria and recruitment of patients.

To address comments from reviewers, we made several changes in the manuscript that clarified methods used and explained implications for the interpretation of results (see answers to comments from reviewers #1 and #2).

2. Please consider performing a sensitivity analysis restricted to countries with large numbers of cases as suggested by reviewer #2 to assess whether results were impacted by the inclusion of countries with few cases; it is likely that the cases from these countries are highly selected and therefore that their inclusion may have led to a selection bias.

In our response to comments from reviewer #2, we have modified the main figures of the manuscript, and only present data from countries with at least 50 observations in each study period. Furthermore, we now report in the *Results section* a sensitivity analysis using mixed-effects logistic regression that only included data from countries meeting this criterion (see answer to reviewer #2 for estimates from this analysis).

Reviewer #1 (Recommendations for the authors):• Figure 2: it is not clear to me why there were 12,665 records excluded from before or after the study periods, as to my mind these would have fit the definition of pre- and post- Omicron. Please explain.

The reason why records of patients admitted to hospital more than two months before or after country-level frequencies of the Omicron variant reached the thresholds used in the definition of the study periods were excluded was to improve comparability between patients who were admitted during the *pre-Omicron period* versus those admitted during the *Omicron period*. For example, patients admitted to hospital several months before the emergence of the Omicron variant most likely differed from patients hospitalised with the Omicron variant not only with regard to the infecting variant but also unmeasured (potential) confounders (e.g. time since last vaccination dose or since previous infection). In epidemiological terms, our objective in restricting the analytical population to relatively narrow time windows was to increase the plausibility of the exchangeability assumption (Greenland and Robins, Identifiability, Exchangeability, and Epidemiological Confounding. International Journal of Epidemiology 1986; VanderWeele, Rothman, Lash, Confounding and Confounders. in Modern Epidemiology. 4^th^ Edition, 2021). We have now included the following statement in the *Methods* section to explain why data from these patients were not analysed (here and throughout this document, changes in the manuscript are underlined):

“For each country, the period during which infections were assumed to be caused by other variants ended in the epidemiological week before the Omicron variant relative frequency crossed a low threshold percentage (e.g., 10%) (see Figure 1). The first epidemiological week when Omicron variant frequency, as a proportion of all circulating variants, was higher than a given threshold percentage (90% in analyses presented in the Results section and 80% in sensitivity analyses) was used as the start date of the period during which all admissions were considered to be caused by the Omicron variant. Note (i) that amongst different countries these two study periods started in different calendar weeks, depending on when the Omicron variant was introduced to the location and on the rate of its local spread, and (ii) that in this analysis all Omicron sub-lineages are included (e.g., BA.1.1, BA.2). Only patients admitted to hospital in the two months before country-level Omicron variant frequency reached the lower threshold and those admitted in the first two months after Omicron variant relative frequency reached 90% were included in the primary analysis; the reason for restricting the study population to those admitted during these time windows was to reduce confounding by unmeasured factors whose frequencies in the hospitalised population also changed over time and which might be associated with clinical outcomes.”

Based on this comment, we also fit, as a sensitivity analysis, a logistic regression model that included the 12,665 records that were excluded from the primary analysis and obtained similar estimates of the association between study period and fatality risk.

• Table S7: please add percentages of patients hospitalized for COVID as well as numbers as these are easier to compare between countries and periods.

We have now included two columns that show percentages for the two study periods.

• Table S11: This analysis shows the results from a standard Cox model where discharge is treated as a censored observation. However, discharge is informative censoring, as presumably, the risk of death is much lower in a discharged patient. The authors may want to consider performing a competing risk analysis (Fine & Grey model) instead of a Cox model for this sensitivity analysis. The results would be more comparable to the logistic regressions, where discharges are included in the denominators but not the numerators.

We would like to thank the reviewer for this suggestion. We agree that a competing risk analysis might be appropriate in this context. We now present results of a competing risk model, fit using the stcrreg command in Stata (https://www.stata.com/manuals/ststcrreg.pdf) (see below). Results of the two different approaches are presented (Latouche et al. A competing risks analysis should report results on all cause-specific hazards and cumulative incidence functions. Journal of Clinical Epidemiology 2013).

“Table S11. Survival models. Results of a Cox proportional hazards model, stratified by country, on time to death in the first 28 days since hospital admission or onset of symptoms, which happened latest, are shown in the Hazard ratio column. For this analysis, if follow-up duration was longer than 28 days, it was set to 28 days, and patients who were discharged were censored on the day of discharge. The assumption of proportional hazards was violated for the variable on previous vaccination; for this reason, the model was also stratified by this variable. An alternative analysis assumed that patients discharged from hospital were censored on day 28; in this analysis, the hazard ratio for the variable corresponding to study period was 0.68 (0.63 – 0.74); for this model, the proportional hazards assumption did not hold for the study period variable. We also fit a competing risk model, with hospital discharge as competing event; estimates from this model are presented in the Subhazard ratio column. In this model, previous COVID-19 vaccination was included as a covariate (subhazard ratio 0.55, 95% CI 0.52 – 0.59). We also fit a competing risk model using only data from the six countries included in Figures 3 – 5 and that included country as a dummy variable; in this model, the subhazard ratio for the Omicron period variable was 0.68 95% CI (0.63 – 0.74).

• Figure S4: it is not clear how vaccination coverage is defined here (1 dose? 2 doses? 3 doses?)

In Figure S4, vaccination coverage refers to any vaccination, i.e. at least one dose. Below is the updated legend of the figure:

“In addition to information on Omicron variant frequency, each panel also shows data on vaccination: the dashed line shows the proportion of population vaccinated with at least one dose relative to the maximum number vaccinated in each country at the time of the analysis (March 2022).”

• The authors may want to comment on the results from individual countries where mortality increased in the Omicron era (ex. India, Brazil, Spain), and the factors that might explain this. Presumably, these results may be due to some confounding by age and clinical profile of patients in both eras.

We have now modified the following paragraph in the *Discussion section* to comment on these results:

“… All these factors might have contributed to the observed association, possibly to different degrees in different countries, reason for which this result should not be assumed to necessarily relate to differences in variant virulence. Of note, data from India (see Figure 5) suggest slightly higher fatality risk during the Omicron period compared to the pre-Omicron period for patients older than 60 years, which could be potentially explained by confounding unrelated to age, residual age-related confounding, not controlled by the categorisation used in our analysis, or alternatively by the limited sample size and consequent uncertainty.”

Note that to address a comment from reviewer #2, regarding presenting data from countries with relatively small study sample sizes, estimates for the Brazilian and Spanish study populations were removed from Figure 5.

Reviewer #2 (Recommendations for the authors):This is a great attempt for collaborative work, and we do know and appreciate how difficult it is to set up such a framework.However, when reading the paper, it looks as if the project is not mature yet. Patients' enrolment per country is extremely disproportional and discredits the project. Can you just include the countries that sent at least 100 patient's data and name all countries as part of the consortium? countries that have sent very few forms should not be mentioned.Patients' data sources seem to differ a lot per country, at the end it looks like a patchwork of random data/patients that is being analyzed to see if it could be of any use.

The reason why numbers of records contributed by different countries are so variable is that datasets from both the United Kingdom and South Africa are part of country-wide, geographically representative epidemiological efforts, whilst in some of the other participating countries, a limited number of ISARIC partner centres recruited patients. As mentioned in another comment by this reviewer (see below), a major strength of this research project is its international and collaborative character, and for this reason, it is important to describe the data contribution of all participating countries. Note that the number of records contributed by each partner institution could be influenced by a multitude of factors: local incidence of SARS-CoV-2 infection during the study period, logistical constraints (e.g. number of staff collecting data), and spread of the Omicron variant locally; the latter determined the distribution of records in the two study periods.

To address this comment, we modified Figures 3, 4 and 5 to include only countries with at least 50 observations in each study period (see updated figures below); figures in the *Supplementary Appendix* presenting country-level data were also updated. Furthermore, in our multivariate analysis of the association between study period and fatality risk, we performed a sensitivity analysis that included this same set of countries; the modified paragraph in the *Results section* is also presented below.

“In a mixed-effects logistic model on 14-day fatality risk that adjusted for sex, age categories, and vaccination status, hospitalisations during the Omicron period were associated with lower risk of death (see Table 2). The inclusion of common comorbidities in the model did not change the estimated association. Similar results were obtained when using 28-day fatality risk as the outcome. We repeated the 14-day fatality risk analysis excluding patients who reported being admitted to hospital due to a medical condition other than COVID-19; the estimated odds ratio for the association between study period and the outcome was similar to those reported in Table 2. In an additional sensitivity analysis, estimates from a model that only included data from countries with at least 50 records per study period were also similar (OR 0.65, 95% CI 0.61 – 0.69, adjusted for covariates included in model I, Table 2). Survival analysis was also performed, and similar results were obtained (Table S11).”

To make the paper clearer I would suggest that you:1. Clarify the goals:If the goal is to assist policy makers when a new variant emerges it will always be too late as variants are not introduced in different countries on the same calendar week. For Omicron (a bad example) policy makers reacted within 24h or less after the first announcement of this new variant.Such collaborative programme could be very useful to identify unique variant characteristics compared to previous variants and identify specific country patterns linked to different variant exposure, different vaccination level, comorbidity rates, access to care etc…such global initiative doesn't exist yet.

We would like to thank this reviewer for the valuable comments and for highlighting the public health value of our work. Global collaborations are needed when responding to international public health threats, and we believe the work reported in this manuscript will motivate research groups to establish similar initiatives. We modified the final paragraph of the manuscript based on this comment:

“In conclusion, we believe our approach of comparing changes in clinical characteristics of COVID-19 using multi-country standardised data, especially when combined with smaller scale studies that collect individual-level data on infecting variants for validation, will be useful in understanding the impact of new variants in the future. Another application will be in using routinely collected health data for cross-country comparisons of variant characteristics. Equally importantly, the successful conduct of this study, and the lessons learned, including the potential weaknesses discussed above, shows that multi-country efforts to study emerging SARS-CoV-2 variants are feasible, improvable and can generate insights to inform policy decision making.”

2. Clarify which patients are included in the data base. All COVID-19 patients hospitalized, only the ones with COVID-19 as main reason for hospitalization, critical care patients, all patients with respiratory symptoms? How are patients with COVID-19 as the main reason for hospitalization?

These questions were addressed in the responses to other comments. Only hospitalised patients with SARS-CoV-2 infection were recruited, and local investigators were responsible for the recruitment procedure. We included the following sentences in the *Discussion section*:

“Another weakness of our study is that recruitment procedure was not standardised and was defined locally. Whilst this likely affected the generalisability of our descriptive estimates (fatality risk and frequencies of symptoms and comorbidities) to local populations of hospitalised COVID-19 cases (Lash and Rothman, Selection Bias and Generalizability. in Modern Epidemiology 4^th^ Edition 2021; Rothman et al. Why representativeness should be avoided. International Journal of Epidemiology 2013), it might not have affected the association between study period and fatality risk, at least not beyond the well-described potential for collider bias in hospital-based studies on COVID-19 outcomes (Griffith et al. Collider bias undermines our understanding of COVID-19 disease risk and severity. Nature Communications 2020).”

3. Standardize the data so that the analysis is not done on different populations for each criteria. Currently symptoms analysis seems to mostly reflect UK patients, vaccination status mostly South Africa patients.

As mentioned in the *Results* and *Discussion section*s, most patients in this study were from the United Kingdom and South Africa. In our analyses, we accounted for this by presenting data stratified by country, including in the main figures of the manuscript. Our regression analysis does include data from all study countries, but variation in risk is accounted for by the statistical method used. We included a statement about this:

“The major strength of our study relates to inclusion of data from all WHO geographic regions, collected with standardised forms, with over 100,000 records. However we note that 96.6% of patients were from two countries – South Africa and the United Kingdom – and that the relative contributions of these countries to the study data were different in the two study periods (Table S5); to avoid misinterpretations linked to changes in country-specific contributions to data in the pre-Omicron and Omicron periods we present descriptive analyses by country and use statistical models that adjust for country-level variation. It is also important to consider the relative contributions of these countries when interpreting descriptive analyses that refer to the combined dataset.”

In addition to including this statement in the manuscript, we have also included a short sub-section in the *Supplementary Appendix* that discusses the frequency of symptoms after excluding data from the United Kingdom:

“Frequency of symptoms outside the United Kingdom and South Africa

Most, 82.5% (N = 579), patients admitted to hospital during the pre-Omicron period outside the United Kingdom and South Africa had at least one symptom; this percentage is lower than the frequency estimated including the United Kingdom data (96.6%), possibly due to the low frequency of symptoms in India (Table S9). The corresponding frequency during the Omicron period was 81.5% (N = 1702).”

4. Again I would keep the top 8 or 10 countries, the small number of data provided by the other countries introduces confusion and alter the impact of your paper.

To address this and another comment from the same reviewer, we have now modified the main figures of the manuscript to include only countries with at least 50 hospitalised patients in each study period. We have also included a sensitivity analysis that estimates odds ratio after excluding countries with limited study data (see above).

Other questions brought up by the reviewer to be addressed by authors (editor's note: originally these questions were in the public review but have been moved to the comments to authors to be addressed):• The study is presented as a multi-center international study that includes more than 100,000 patients from 30 countries, however, 96.6% of the study patients originated from 2 countries, South Africa (54%) and the United Kingdom (42.6%). Can this still qualify as a multicenter study?Response to comment

Eight countries contributed at least 100 records during the two study periods, and three additional countries recruited more than 100 patients but due to the timing of the local spread of the Omicron variant some of these patients were not included in the analysis. We believe this justifies referring to this study as an international, or multi-country, epidemiological study. It is also important to note that data from multiple countries, including from the United Kingdom and South Africa, were collected in multiple centres.

• Country specific medians suggest that the younger age of patients after the Omicron variant experience in the combined dataset is at least partially explained by an increase of data contributed by South Africa. Could the high proportion of South African patients' data have impacted on other measures too?

As there is considerable between-country variation in vaccination coverage, we did not report aggregated frequencies for the history of vaccination, and most of the statements in the *Results* sub-section *Vaccination history in hospitalised patients* refer to Figure S3 and Table 1, which present data stratified by country. For this reason, we believe that data from South Africa did not impact reporting of the vaccination outcome. Regarding the fatality risk, our multivariate analysis uses random intercepts that account for differences in risk between countries; furthermore, our analysis is adjusted for factors that might vary in distribution between countries, such as age and history of vaccination.

• Are the patients recruited COVID-19 proven patients? Are incidental cases included or excluded? How was the primary reason for hospitalization identified? Interpretation of mortality rate could be influenced by the recruitment of patients.

As described in the *Results section*, patients with negative PCR test result for SARS-CoV-2 detection were excluded from the analysis. The statement is the following:

“All patients from South Africa, the United Kingdom and Malaysia were assumed to be SARS-CoV-2 positive, as this is one criterion for inclusion in their databases. Of the 2,296 records from other countries, information on SARS-CoV-2 diagnostic testing was available for 1,999 observations; whilst patients with negative PCR test result (N=10) were excluded from the rest of the analysis, those with missing PCR data (N=297) were assumed positive (see Table S5 for distribution by country).”

Furthermore, information on the primary reason for hospitalisation was available for nearly 30,000 patients; ~70% reported that the reason for hospitalisation was COVID-19. In the *Discussion section*, we mention that incidental infections might have affected our findings, although the inclusion of patients with incidental infections in hospital-based COVID-19 epidemiological studies is common and not specific to our design. The following paragraph was modified:

“One possible explanation for this finding would be if incidental SARS-CoV-2 infections, i.e. infections that were not the primary reason for hospitalisation, were more frequent during the Omicron period; the high transmissibility of this variant, and the consequent peaks in numbers of infections, together with its reported association with lower severity, provides support for this hypothesis. However, in the subset of patients with data on the reason for hospitalisation there was no increase in the proportion of admissions thought to be incidental infections and indeed proportions in both study periods were consistent frequencies of incidental infections in recent studies in the United States (Klann et al. Distinguishing Admissions Specifically for COVID-19 From Incidental SARS-CoV-2 Admissions: National Retrospective Electronic Health Record Study. J Med Internet Res) and the Netherlands (Voor in ’t holt et al. Admissions to a large tertiary care hospital and Omicron BA.1 and BA.2 SARS-CoV-2 polymerase chain reaction positivity: primary, contributing, or incidental COVID-19. International Journal of Infectious Diseases 2022), although in the latter, non-incidental infections included patients for whom COVID-19 was a contributing but not the main cause of hospitalisation.”

Reviewer #3 (Recommendations for the authors):I only have one specific suggestion for further analysis:In the Results section on vaccination history, it is implied that more granular detail on vaccination status of patients (ie number of vaccines received) is available but is not used in the models. If this is the case, it might be interesting to see a version of the main model adjusted on number of vaccine doses, even if this is restricted to countries where this data is available.

We performed an analysis that adjusted for the number of vaccine doses received; 19,360 patients, 18.8% of the study population, were included. In the model, also adjusted for sex and age, the odds ratio quantifying the association between study period and fatality risk is 0.72 95% confidence interval (0.63 – 0.82)**.**